# Peripheral glia and neurons jointly regulate activity-induced synaptic remodeling at the *Drosophila* neuromuscular junction

**Yen-Ching Chang[1], Yi-Jheng Peng[1], Joo Yeun Lee[1†], Annie Wen[1], Karen T Chang[1,2]***

[1]Zilkha Neurogenetic Institute, Keck School of Medicine, University of Southern California, Los Angeles, United States; [2]Department of Physiology and Neuroscience, Keck School of Medicine, University of Southern California, Los Angeles, United States

***For correspondence:**
changkt@usc.edu

**Present address:** †Department of Ophthalmology, University of California, San Francisco, United States

**Competing interest:** The authors declare that no competing interests exist.

## eLife Assessment

This study presents a **valuable** finding on a new role of glia in activity-dependent synaptic remodeling using the Drosophila NMJ as a model system. The evidence supporting the claims of the authors is **convincing**. The authors have addressed most of the reviewers' concerns and help to further clarify the claims. The work will be of interest to neuroscientists working on glia-neuron interaction and synaptic remodeling.

**Abstract** In the nervous system, reliable communication depends on the ability of neurons to adaptively remodel their synaptic structure and function in response to changes in neuronal activity. While neurons are the main drivers of synaptic plasticity, glial cells are increasingly recognized for their roles as active modulators. However, the underlying molecular mechanisms remain unclear. Here, using *Drosophila* neuromuscular junction (NMJ) as a model system for a tripartite synapse, we show that peripheral glial cells collaborate with neurons at the NMJ to regulate activity-induced synaptic remodeling, in part through a protein called shriveled (Shv). Shv is an activator of integrin signaling previously shown to be released by neurons during intense stimulation at the fly NMJ to regulate activity-induced synaptic remodeling. We demonstrate that Shv is also present in peripheral glia, and glial Shv is both necessary and sufficient for synaptic remodeling. However, unlike neuronal Shv, glial Shv does not activate integrin signaling at the NMJ. Instead, it regulates synaptic plasticity in two ways: (1) maintaining the extracellular balance of neuronal Shv proteins to regulate integrin signaling, and (2) controlling ambient extracellular glutamate concentration to regulate postsynaptic glutamate receptor abundance. Loss of glial cells showed the same phenotype as loss of Shv in glia. Together, these results reveal that neurons and glial cells homeostatically regulate extracellular Shv protein levels to control activity-induced synaptic remodeling. Additionally, peripheral glia maintain postsynaptic glutamate receptor abundance and contribute to activity-induced synaptic remodeling by regulating ambient glutamate concentration at the fly NMJ.

## Introduction

The nervous system is highly plastic, with the capacity to undergo dynamic alterations in structure and strength in response to changing stimuli and environments (*Lamprecht and LeDoux, 2004*; *Citri and Malenka, 2008*). This activity-induced synaptic remodeling process is highly conserved across

species and plays crucial roles in circuit formation during development, as well as in stabilizing existing connections post-development. Synaptic remodeling represents a finely orchestrated process, with communications across both the pre- and postsynapses to allow the growth of new synapses, and the stabilization and strengthening of existing ones. While much of the research on synaptic plasticity has concentrated on interactions between presynaptic axon terminals and postsynaptic cells, most synapses are tripartite synapses, with glial cells as the third cell type (*Araque and Navarrete, 2010*). Beyond simply providing metabolic support, glial cells are increasingly recognized for their roles as active modulators of synaptic plasticity (*Sancho et al., 2021*). However, the mechanisms by which glial cells and neurons collaborate to coordinate activity-induced synaptic remodeling are not well understood.

The *Drosophila* larval neuromuscular junction (NMJ) is a genetically tractable system and a tripartite glutamatergic synapse that serves as an excellent model system to investigate mechanisms underlying activity-induced synaptic remodeling by glial cells (*Banerjee and Bhat, 2008*; *Freeman, 2015*; *Kim et al., 2020*). The peripheral glial cells at the fly NMJ perform some of the key functions similar to mammalian glia, including controlling neuronal excitability and conduction velocity (*Kottmeier et al., 2020*; *Rey et al., 2023*), recycling of neurotransmitters (*Rival et al., 2004*; *Danjo et al., 2011*), and engulfing and clearing debris during damage to allow the growth of new boutons (*Fuentes-Medel et al., 2009*). They also release proteins such as transforming growth factor (TGF-β) to support synaptic growth (*Fuentes-Medel et al., 2012*), TNF-α (eiger) to influence neuronal survival (*Keller et al., 2011*), Wingless to regulate GluR clustering (*Kerr et al., 2014*), and laminin to control animal locomotion (*Petley-Ragan et al., 2016*). Nevertheless, the role of peripheral glia in regulating activity-induced stabilization and remodeling of the existing synapses at the NMJ remains unknown.

Previous studies on activity-induced synaptic remodeling at the fly NMJ demonstrated that neuronal activity leads to the enlargement of existing boutons, accompanied by increases in postsynaptic GluR abundance (*Lee et al., 2017*; *Chang et al., 2024*). Intense neuronal stimulation triggers the release of a protein called Shriveled (Shv) by presynaptic motoneurons, which activates βPS integrin bi-directionally to stimulate synaptic bouton enlargement and elevate GluR levels on the postsynaptic muscles (*Lee et al., 2017*). Consequently, *shv* mutants display defective post-tetanic potentiation (PTP), a form of functional synaptic plasticity similar to the early phase of long-term potentiation (LTP) seen in mammalian neurons. Here, we demonstrate that the Shv protein is also expressed in glial cells and is released extracellularly by peripheral glial cells. Glial Shv not only regulates basal GluR clustering but is also required for activity-induced synaptic remodeling. We further demonstrate that while glial Shv is present extracellularly, it does not respond to neuronal activity, nor does it activate integrin signaling, unlike Shv derived from neurons. Instead, glial Shv contributes to synaptic plasticity regulation by modulating the levels of Shv release from neurons and by controlling the levels of ambient glutamate concentration. Restoring ambient glutamate concentration could correct basal GluR abundance and defective synaptic plasticity caused by the loss of glial cells. These results further reveal that regulation of ambient extracellular glutamate concentration by glia is an important mechanism contributing to synaptic plasticity regulation.

## Results

### Shv is expressed in peripheral glia

To determine the role of Shv in glia, we first monitored its presence in different cell types. To this end, we knocked in eGFP to the 3'-end of the full-length Shv protein using CRISPR/Cas9-catalyzed homology-directed repair (HDR; *Gratz et al., 2014*). Western blots confirmed that the Shv protein is tagged with eGFP (*Figure 1A*), and immunostaining revealed its presence in neurons and glial cells (*Figure 1B*). Glial cells were identified using antibody against reverse polarity (Repo), a transcription factor expressed exclusively in glial cells (*Xiong et al., 1994*), and neuronal cells were marked either by Elav, a transcription factor expressed in neurons (*Robinow and White, 1991*), or HRP, which stains the neuronal cell membrane. We found that Shv-eGFP is present in both Repo and Elav positive cells in the larval brain, consistent with our previous report (*Lee et al., 2017*). Furthermore, Shv-eGFP can be detected at the NMJ, with weak signals in postsynaptic muscles and synaptic boutons, and stronger signals in peripheral glia (*Figure 1C*).

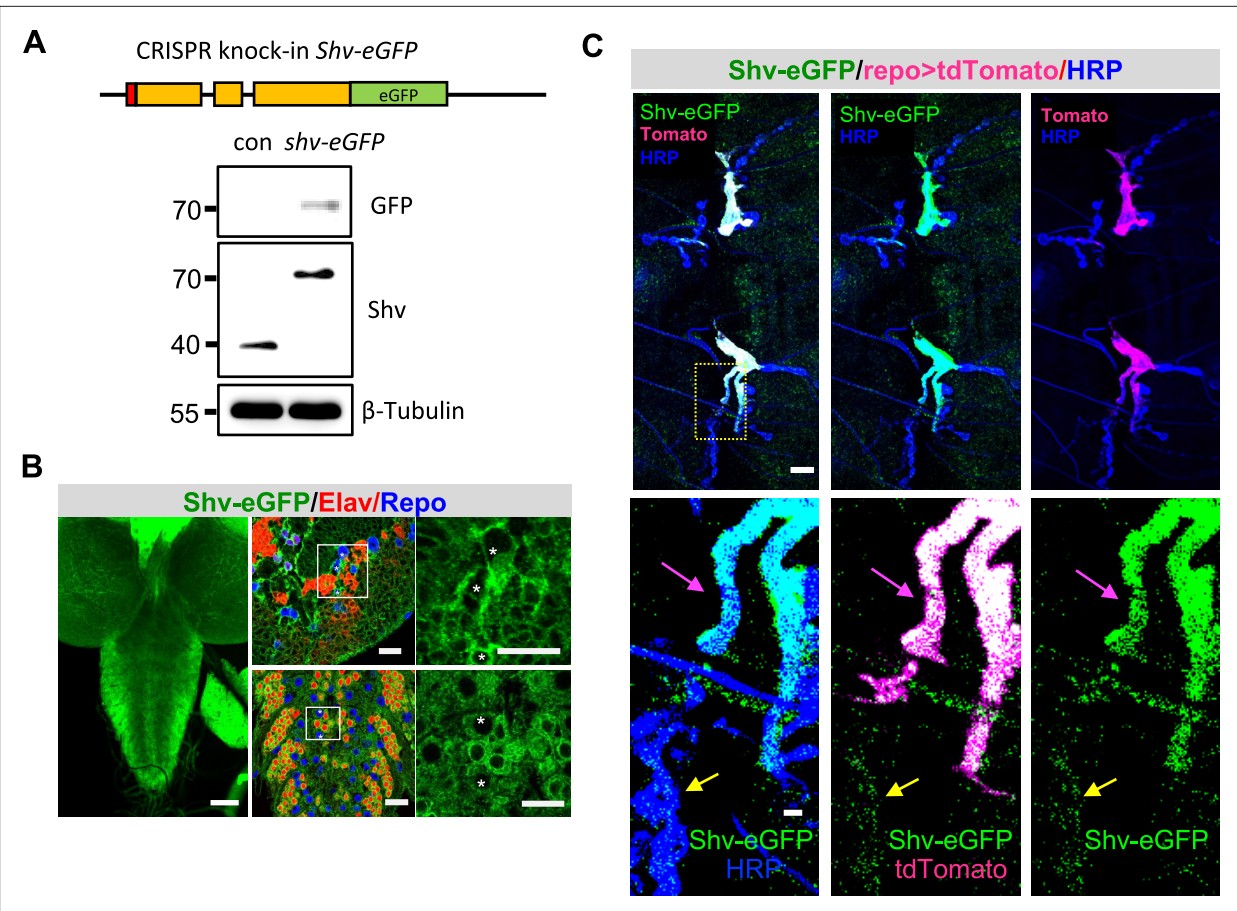

**Figure 1.** Endogenous Shv tagged with eGFP is detected in glial cells. (**A**) Schematic of eGFP knock-in to the Shv gene generated using CRISPR/Cas9 system (top). Exons are in orange, signal peptide in red. Western blots using antibodies against GFP and Shv confirm the presence of eGFP in Shv. β-Tubulin is used as a loading control. (**B**) Low magnification images of the third-instar larval brain showing weak Shv-eGFP signal throughout the brain (left). Scale bar = 50 μm. Zoomed-in view of the brain hemisphere and ventral nerve cord with neurons and glia marked by Elav and Repo antibodies, respectively. Asterisks label glial cells with Shv expression. Scale bar for brain and VNC are 10 and 15 μm, respectively. (**C**) Shv-eGFP can be detected at the neuromuscular junction (NMJ). Glial membrane is marked by membrane targeted tdTomato (driven by glial specific *repo-GAL4*) and neuronal membrane labeled by HRP. Zoomed-in views show that Shv-eGFP colocalizes with glial membrane (magenta arrow) and synaptic boutons (yellow arrow). Shv-eGFP also weakly labels the muscle. Note that a single optical slice of the NMJ at muscle 6/7, abdominal segment 2 is shown, which highlights Shv-eGFP colocalization with glia and synaptic boutons in this permeabilized prep. The full glial stalk is not visible because it lies in a different focal plane from the branch of interest. Scale bar = 10 μm in the upper panels, and 2 μm in the lower panels.

The online version of this article includes the following source data for figure 1:

**Source data 1.** PDF file containing original western blots for *Figure 1A*, showing the relevant bands and molecular weight marker.

**Source data 2.** Original files for western blot analysis are shown in *Figure 1A*.

## Glial Shv is required for activity-induced synaptic remodeling

Loss of Shv in the *shv¹* mutant was previously shown to be essential for activity-induced synaptic remodeling (*Lee et al., 2017*). Given that Shv is observed in neurons, glia, and muscle (*Figure 1C*), we determined the tissue-specific requirement for Shv by systematically knocking down *shv* using RNAi and tissue-specific drivers (*Figure 2A*). We found that *shv* knockdown in neurons using the pan-neuronal driver, *elav-GAL4*, generated the same phenotype as *shv¹* mutant (*Lee et al., 2017*), namely smaller bouton size, reduced basal GluR intensity, and defective synaptic remodeling in response to neuronal stimulation. However, knockdown of *shv* in glia using the pan-glial specific driver, *repo-GAL4*, resulted in normal bouton size but significantly elevated GluR levels, as well as abolished activity-induced synaptic remodeling. Knockdown of *shv* in glia using another independent RNAi line resulted in the same phenotypes (*Figure 2—figure supplement 1A*). This increase in GluR is unexpected and further suggests that neuronal and glial Shv acts through different pathways

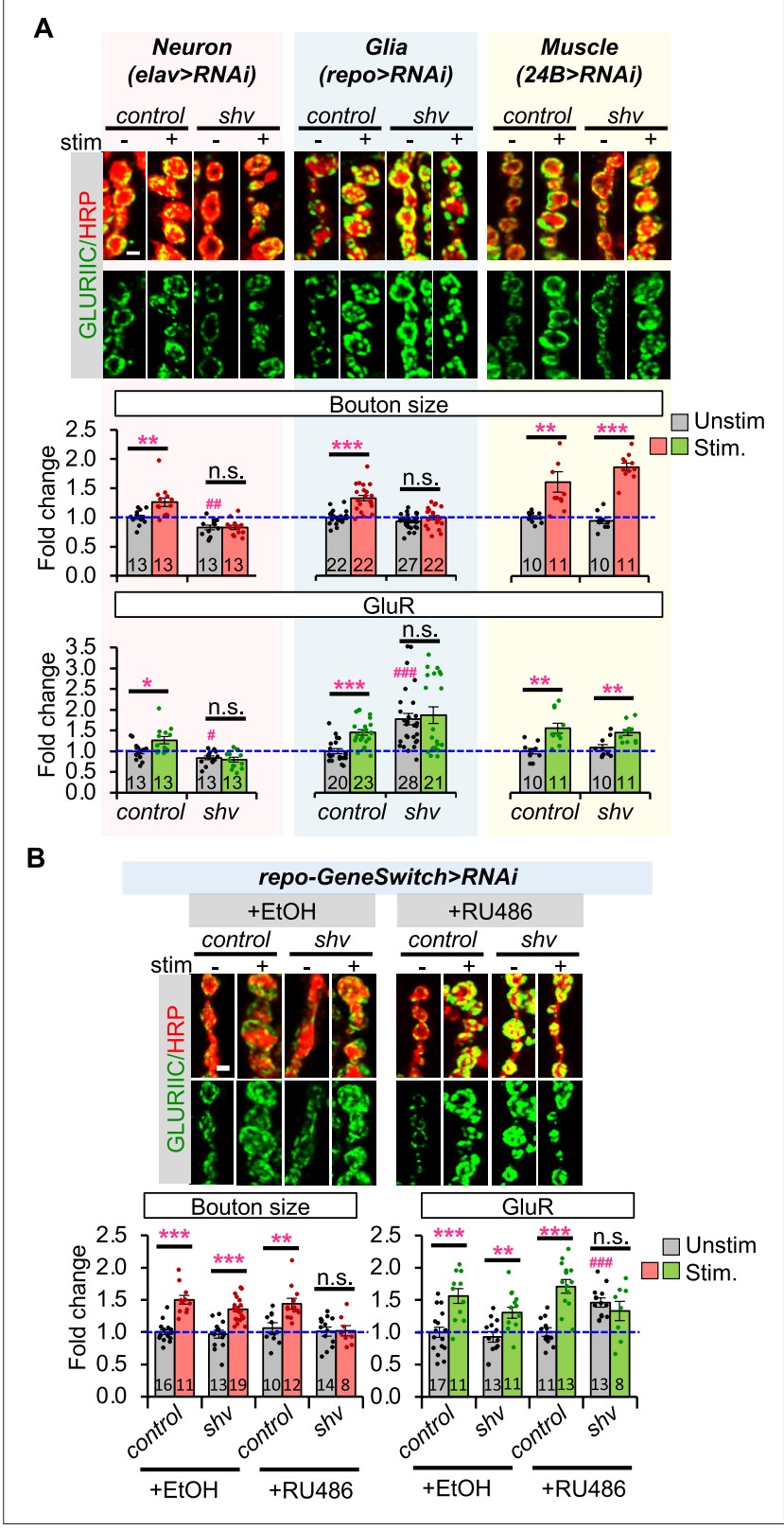

**Figure 2.** Shv in glia is necessary for activity-induced synaptic remodeling. (**A**) Tissue-specific knockdown of *shv*. Reducing Shv in neurons or glia blocks activity-induced synaptic remodeling, but not when *shv* is knocked down in muscles. Notably, glia-specific knockdown of *shv* increases the levels of GluRIIC, whereas neuronal Shv knockdown decreases basal GluRIIC. (**B**) Acute knockdown of *shv* in glia using the inducible *repo-GeneSwitch-GAL4* driver

*Figure 2 continued on next page*

*Figure 2 continued*

affects basal GluR intensity and abolishes activity-induced synaptic changes. (**A, B**) Scale bar = 2 μm. All values are normalized to unstimulated control and presented as mean ± SEM. Control contains TRiP RNAi control vector driven by the indicated driver. Statistics: One-way ANOVA followed by Tukey's multiple comparison test was used to compare between unstimulated control and unstimulated neuromuscular junctions (NMJs) across genotypes. Student's *t*-test was used to compare between unstimulated and stimulated NMJs of the same genotype. #p ≤ 0.05; ##p ≤ 0.01; ###p ≤ 0.001 when comparing unstimulated samples to unstimulated control. *p ≤ 0.05; **p ≤ 0.01; ***p ≤ 0.001 when comparing stimulated to unstimulated NMJs.

The online version of this article includes the following source data and figure supplement(s) for figure 2:

**Source data 1.** Data for relative bouton size and GluR intensity.

**Figure supplement 1.** Independent *Shv-RNAi* line and efficacy of *shv* knockdown by the GeneSwitch system.

**Figure supplement 1—source data 1.** Raw data for relative bouton size, GluR intensity, and protein levels.

**Figure supplement 1—source data 2.** PDF file containing original western blots, showing the relevant bands and molecular weight marker.

**Figure supplement 1—source data 3.** Original files for western blots.

to regulate GluR abundance. Lastly, knockdown of *shv* in postsynaptic muscles using the muscle-specific driver *24B-GAL4* did not affect synaptic development or activity-induced synaptic changes. Collectively, these results are consistent with an earlier finding that Shv is predominantly released by neurons to maintain activity-induced synaptic remodeling (*Lee et al., 2017*), as well as reveal an unexpected requirement for glial Shv in sustaining activity-induced synaptic changes and basal GluR abundance.

Given that reducing Shv in glia altered basal GluR levels during development, we used the GeneSwitch system to determine the temporal requirement for Shv in glia (*Figure 2B*). In the presence of RU486, the *repo-GeneSwitch* driver can undergo a conformational change to activate gene expression in glia (*Osterwalder et al., 2001*; *Roman et al., 2001*; *Artiushin et al., 2018*). First, we confirmed the efficiency of acute *shv* knockdown by performing western blot analysis of dissected larval brains (*Figure 2—figure supplement 1B*). Acute glial knockdown using the *repo-GeneSwitch* driver resulted in a 30% reduction in Shv levels (+RU486), similar to the decrease observed with the *repo-GAL4* driver, suggesting that the GeneSwitch driver is functional. Furthermore, knockdown of *shv* by the ubiquitous *tubulin-GAL4* driver completely eliminated Shv protein, indicating that the RNAi construct is effective. We next examined the effect of acute glial *shv* knockdown on synaptic remodeling. As shown in *Figure 2B*, transient glial *shv* knockdown is sufficient to elevate basal GluR levels and abolish activity-induced synaptic remodeling. Taken together, these results suggest that while glial Shv is a minor source, it is acutely required for regulating GluR abundance and synaptic plasticity.

Next, we determined the spatial requirement for Shv in glial cells. The peripheral glia can be divided into three subtypes: wrapping glia (WG) is the innermost layer, wrapping and contacting the peripheral nerve bundle; the subperineurial glial (SPG) covers the WG, establishing the blood–brain barrier; the perineurial glia (PGs) is located on the outermost surface of the nerve that is also part of the blood–brain barrier (*Awasaki et al., 2008*; *Stork et al., 2008*; *Brink et al., 2012*; *Freeman, 2015*; *Fernandes et al., 2024*; *Figure 3A*). Using GAL4 lines previously shown to drive expression in specific glial subtypes (*Stork et al., 2012*), we found that reducing Shv in either SPG or PG was sufficient to block activity-induced synaptic remodeling (*Figure 3B*), phenocopying *shv* knockdown in all glial cells (*Figure 2A*). However, we noticed that knockdown of *shv* in SPG more closely resembles the pan-glial knockdown phenotype, likely because SPG, as the middle glial cell layer in the fly peripheral nervous system, may also influence the PG layer through signaling mechanisms (*Lavery et al., 2007*). Conversely, NMJs with *shv* knockdown in WG (*nrv-Gal4*) exhibited normal activity-induced synaptic remodeling (*Figure 3B*). We also tested whether the astrocyte-like glia located in the central nervous system also contributes by knocking down Shv using *alrm-GAL4* (*Doherty et al., 2009*). We observed normal activity-induced synaptic remodeling (*Figure 3*). Together, these results suggest that Shv in PG and SPG glia, both in contact with synaptic boutons, extending into the muscles, and are part of the blood–brain barrier, are crucial for Shv function in glia and for synaptic remodeling.

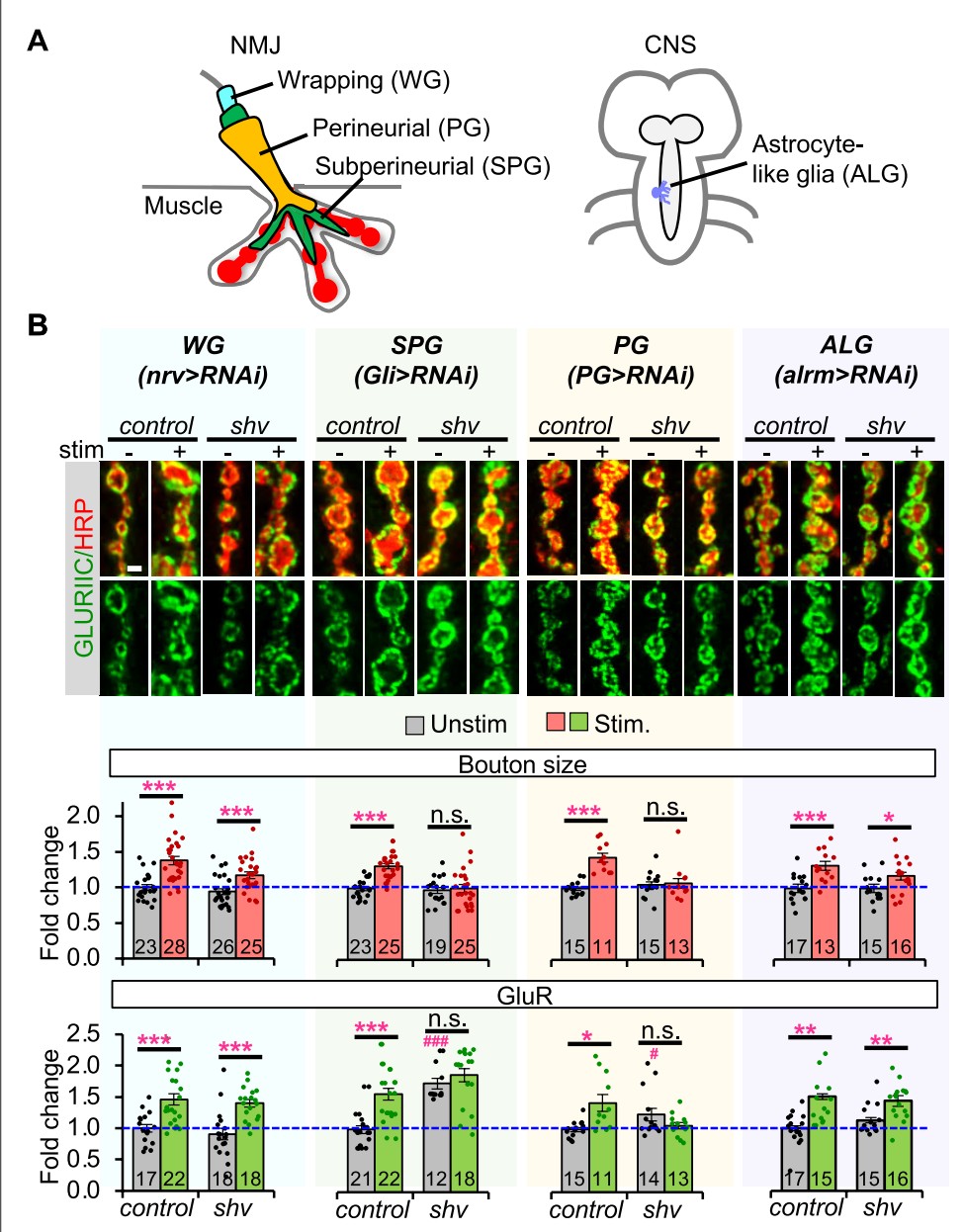

**Figure 3.** Expression of *shv* in perineurial (PG) and subperineurial glia (SPG) is required for activity-induced synaptic remodeling. (**A**) Diagram of relative membrane position and extension of wrapping glia (WG), SPG, and PG at the neuromuscular junction (NMJ). Astrocyte-like glia can be detected in the central brain and ventral nerve chord. (**B**) Representative images and quantification of synaptic changes following knockdown of *shv* in glia subtypes. Knockdown of *shv* in SPG and PG recapitulates the phenotypes of pan-glial knockdown, as well as abolishes activity-induced synaptic remodeling. Scale bar = 2 μm. All values are normalized to unstimulated control and presented as mean ± SEM. Control contains TRiP RNAi control vector driven by the indicated driver. Statistics: One-way ANOVA followed by Tukey's multiple comparison test was used to compare between unstimulated control and unstimulated NMJs across genotypes. Student's *t*-test was used to compare between unstimulated and stimulated NMJs of the same genotype. #$p \leq 0.05$; ###$p \leq 0.001$ when comparing unstimulated samples to unstimulated control. *$p \leq 0.05$; **$p \leq 0.01$; ***$p \leq 0.001$ when comparing stimulated to unstimulated NMJs.

The online version of this article includes the following source data for figure 3:

**Source data 1.** Data for relative bouton size and GluR intensity shown in *Figure 3B*.

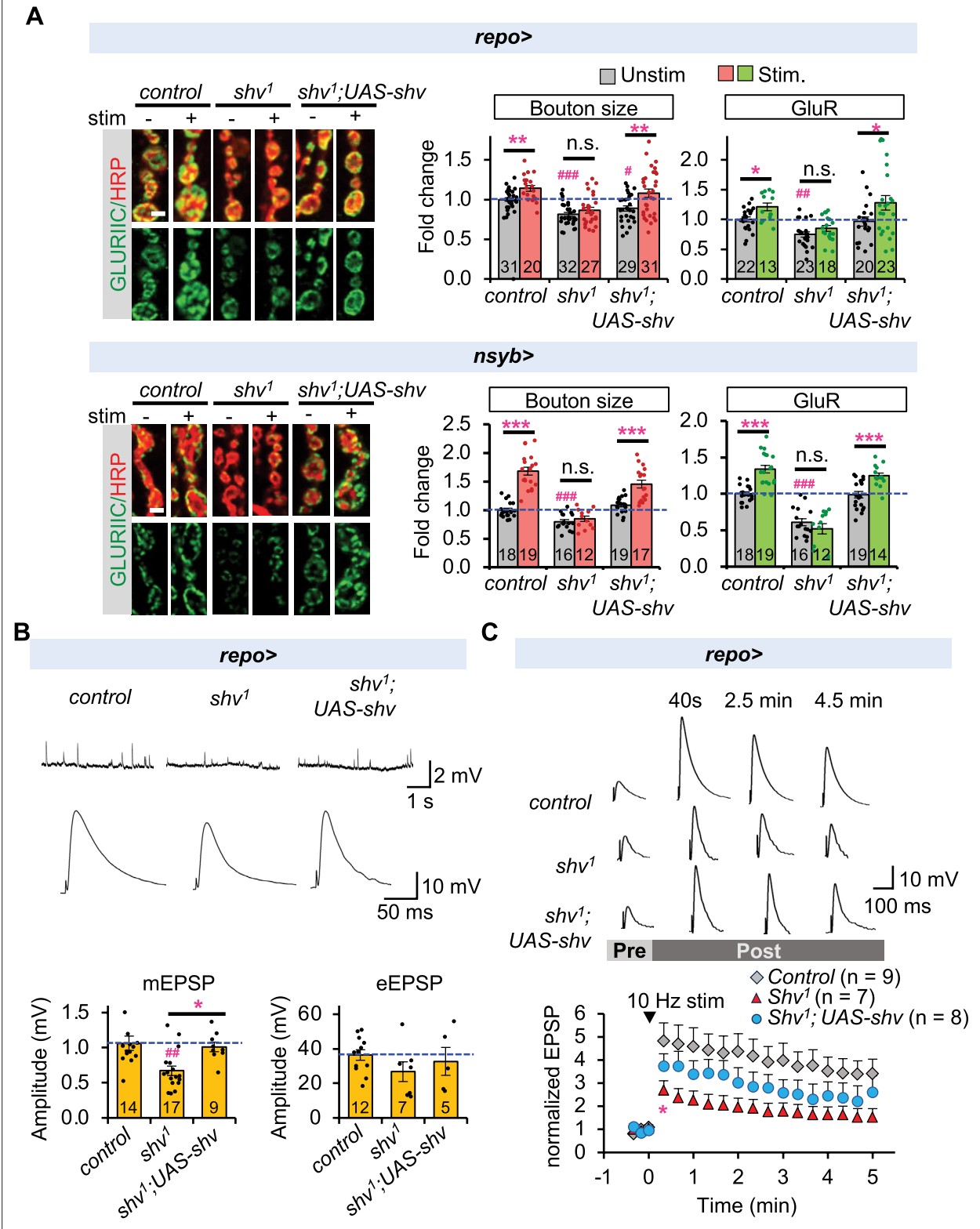

**Figure 4.** Glial Shv rescues defective synaptic plasticity observed in *shv¹* mutant. (**A**) Selective expression of *shv* in glia or neurons of *shv¹* mutants is sufficient to rescue activity-dependent synaptic changes, but glial *shv* expression did not restore basal bouton size. Scale bar = 2 μm. (**B**) Representative mEPSP and eEPSP recordings conducted using HL3 solution containing 0.5 mM Ca²⁺. Average eEPSP amplitude is plotted after nonlinear summation correction. (**C**) Normalized eEPSP before and following tetanus (10 Hz for 2 min) at the indicated time points. Recordings were done using HL3 solution containing 0.25 mM Ca²⁺. *shv¹* showed significantly diminished post-tetanic potentiation (PTP) following stimulation. Expression of *shv* in glial cells of

*Figure 4 continued on next page*

*Figure 4 continued*

*shv*[1] rescued PTP. The number of neuromuscular junctions (NMJs) examined is shown in parentheses. Student's *t*-test was used to compare between control and the indicated genotypes. * shows that *shv*[1] displays PTP that is significantly lower than the control (p ≤ 0.05), starting from the indicated time and onwards. All values are mean ± SEM. For (**A**) and (**B**), one-way ANOVA followed by Tukey's multiple comparison test was used to compare between unstimulated control and unstimulated NMJs across genotypes. Student's *t*-test was used to compare between unstimulated and stimulated NMJs of the same genotype. #p ≤ 0.05; ##p ≤ 0.01; ###p ≤ 0.001 when comparing unstimulated samples to unstimulated control. *p ≤ 0.05; **p ≤ 0.01; ***p ≤ 0.001 when comparing stimulated to unstimulated NMJs.

The online version of this article includes the following source data for figure 4:

**Source data 1.** Data for relative bouton size and GluR intensity, and electrophysiology data.

## Glial expression of Shv is sufficient to rescue synaptic plasticity in *shv* mutants

To determine whether Shv from either neuron or glia is sufficient for activity-induced synaptic remodeling, we expressed *shv* (*UAS-Shv*) in either cell type in *shv*[1] mutants, which exhibit defective synaptic remodeling (*Lee et al., 2017*). As shown in *Figure 4A*, *shv* expression in either neurons or glia was sufficient to restore basal GluR intensity and activity-induced synaptic remodeling; however, glial expression did not normalize bouton size. These data demonstrate that while neuronal Shv is necessary to regulate bouton size during development and modulate synaptic plasticity (*Figure 4A*), Shv derived from glia is also sufficient to maintain basal GluR levels and support activity-induced synaptic modifications.

We have previously shown that neuronal Shv is sufficient to rescue functional plasticity (*Lee et al., 2017*), but the role of glial Shv in the process is not known. Electrophysiological recordings demonstrated that *shv*[1] mutants displayed reduced miniature excitatory potential (mEPSP) amplitude but normal evoked EPSP (*Figure 4B*), consistent with a previous report (*Lee et al., 2017*). Selective expression of *shv* in glia restored mEPSP amplitude, in line with the rescue in GluR levels. Glial-specific expression of *shv* also rescued PTP, an activity-dependent plasticity in *Drosophila* that is functionally similar to the initial stages of LTP (*Figure 4C*). Collectively, these data suggest that glial Shv is sufficient to support functional and structural plasticity.

## Glial Shv does not activate integrin signaling

Shv was previously shown to be released by neurons to trigger synaptic remodeling through integrin activation (*Lee et al., 2017*); we thus asked whether glial Shv restores synaptic plasticity in *shv*[1] mutants through the same mechanism. *Figure 5A* shows that glia can indeed release Shv, as glial expression of HA-tagged Shv (driven by *repo-GAL4*) can be detected extracellularly when stained using non-permeabilizing conditions. Additionally, western blot analysis showed that either neuronal or glial expression of Shv-HA resulted in a protein of the same molecular weight (*Figure 5—figure supplement 1A*), suggesting that major differences in protein processing are unlikely. Next, we assessed the effects of glial Shv on integrin signaling by monitoring the levels of phosphorylated focal adhesion kinase (pFAK), as its levels strongly correlate with integrin activation (*Mitra et al., 2005*; *Tsai et al., 2008*). To our surprise, glial expression of *shv* in *shv*[1] mutant did not restore pFAK levels to normal (*Figure 5B*). Furthermore, *shv* knockdown in glial cells exhibited higher pFAK staining compared to the control (*Figure 5C*), a result that is opposite to what was observed with shv knockdown in neurons (*Lee et al., 2017*). *shv* overexpression in glia, although did not change basal pFAK levels, blocked the activity-induced pFAK increases normally seen in control (*Figure 5C*). Collectively, these results suggest that glial Shv does not activate integrin and appears to play an inhibitory role in activity-induced integrin activation.

To elucidate whether glial Shv directly inhibits activity-induced integrin activation, we monitored the amount of Shv released by glia during neuronal activity by expressing Shv-HA in either glia or neurons of *shv*[1] and measured its extracellular levels. *Figure 5D* demonstrates that while stimulation significantly increased extracellular Shv released by neurons, the amount of extracellular Shv released by glial cells was reduced. This activity-induced decrease in glial Shv levels, along with reduced integrin activation (*Figure 5B*), suggests that glial Shv does not act by directly inhibiting integrin signaling. We further investigated whether the decrease in extracellular glial Shv results from altered release. To address this, we monitored intracellular Shv levels using a permeabilized preparation (we found that

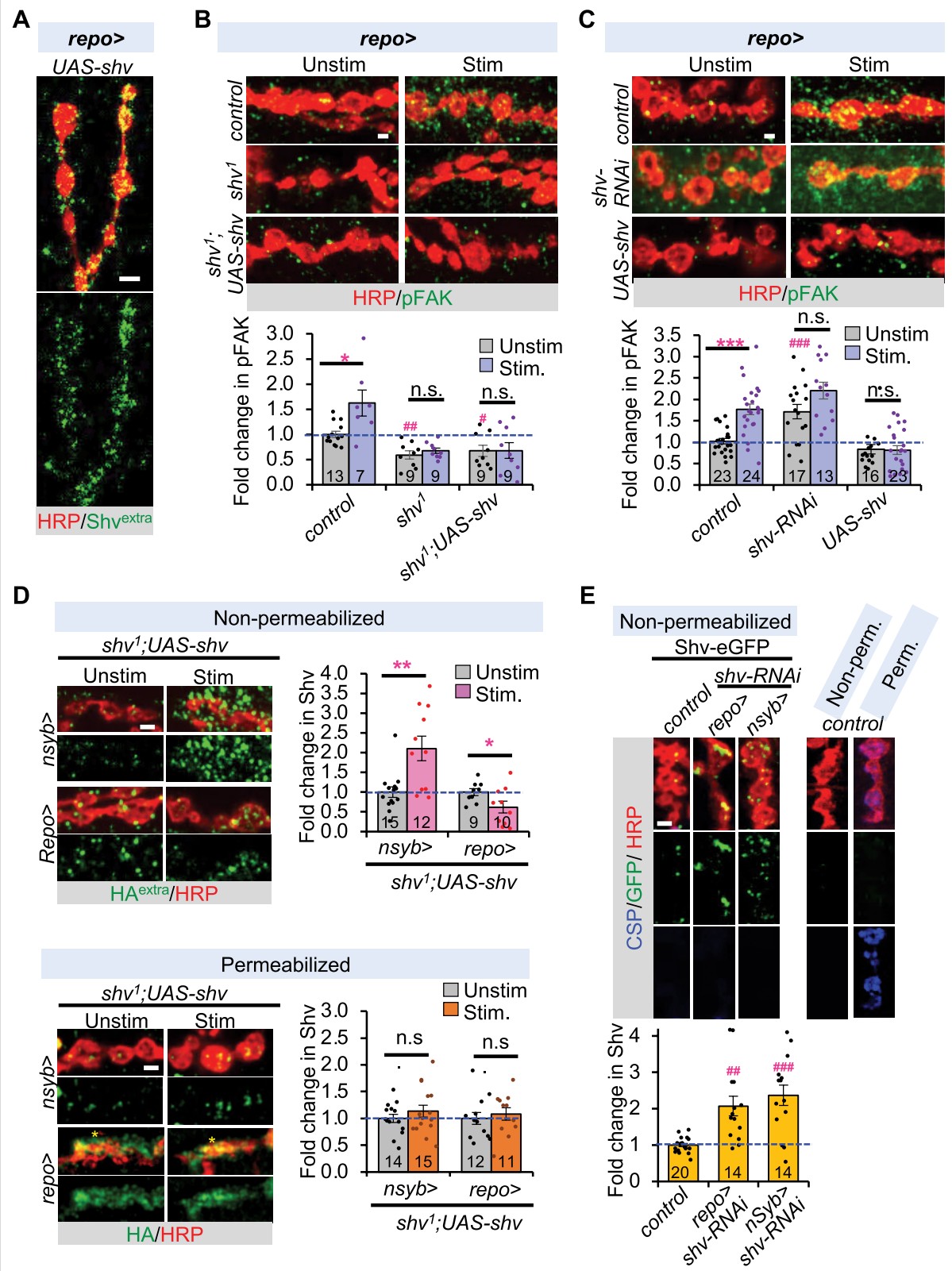

**Figure 5.** Glial Shv does not activate integrin signaling but modulates neuronal release of Shv. (**A**) Representative images showing that Shv can be detected extracellularly when expressed using glial-specific driver. Extracellular Shv-HA (Shv^extra) is monitored using an antibody against HA under detergent-free staining condition. (**B**) Expression of Shv using the glial-specific driver does not rescue pFAK levels during unstimulated and stimulated conditions, revealing that glial Shv does not activate integrin. (**C**) Knockdown of *shv* in glia upregulated pFAK, whereas upregulation of *shv* in glia

*Figure 5 continued on next page*

*Figure 5 continued*

blocked the activity-dependent increase normally seen in control. (**D**) Expression of *shv* in *shv¹* mutants shows activity-dependent release of Shv by neurons, but not by glia. HA^extra indicates extracellular Shv stained under non-permeabilizing (detergent-free) conditions. Permeabilized staining protocol washes away extracellular staining and thus mainly detects intracellular Shv. Intracellular levels of Shv in neurons or glia did not change following stimulation. Asterisk indicates glial membrane overlay with neurite at the neuromuscular junction (NMJ). (**E**) Knockdown of endogenous Shv-eGFP in neurons or glia did not diminish extracellular presence of Shv-eGFP at the NMJ, suggesting homeostatic regulation of Shv protein level. Right panels show control NMJ (*nsyb-GAL4/+*) stained using non-permeabilizing (Non-perm) and permeabilizing (Perm) conditions. CSP, an intracellular protein, is only observed when the sample is stained under permeabilizing conditions, suggesting that our non-permeabilizing protocol is selective for extracellular proteins. Scale bar = 2 µm for all panels. All values are normalized to unstimulated control and presented as mean ± SEM. One-way ANOVA followed by Tukey's multiple comparison test was used to compare between unstimulated control and unstimulated NMJs across genotypes. Student's *t*-test was used to compare between unstimulated and stimulated NMJs of the same genotype. #p ≤ 0.05; ##p ≤ 0.01; ###p ≤ 0.001 when comparing unstimulated samples to unstimulated control. *p ≤ 0.05; **p ≤ 0.01; ***p ≤ 0.001 when comparing stimulated to unstimulated NMJs.

The online version of this article includes the following source data and figure supplement(s) for figure 5:

**Source data 1.** Relative staining intensities for the indicated antibodies and conditions.

**Figure supplement 1.** Neuronal and glial expression of Shv and efficiency of *shv* knockdown by RNAi in neurons and glia.

**Figure supplement 1—source data 1.** PDF file containing original western blots, showing the relevant bands and molecular weight marker.

**Figure supplement 1—source data 2.** Original files for western blots.

**Figure supplement 2.** Integrity of peripheral glial cell membranes.

detergent treatment stripped away extracellular Shv signal). When combined with non-permeabilized extracellular staining, this approach provides insights into total Shv levels. As shown in *Figure 5D*, there was no intracellular accumulation of Shv and the intracellular levels remained unchanged following stimulation, suggesting that reduced extracellular glial Shv is unlikely due to defects in the release machinery.

Next, we tested the hypothesis that glial Shv instead influences integrin signaling by regulating Shv release from neurons. To this end, we monitored Shv secreted by neurons or glia using the endogenously tagged Shv-eGFP line and tissue-specific knockdown of *shv*. When *shv* was selectively knocked down in glia, an increase in extracellular Shv-eGFP level was observed at the NMJ (*Figure 5E*), which likely originated from neurons. This finding is consistent with a role for glial Shv in suppressing neuronal Shv release and further explains the higher pFAK level observed in the case of glial *shv* knockdown (*Figure 5C*). A similar compensatory upregulation from glia was obtained when Shv was knocked down in neurons (*Figure 5E*). To validate that our staining protocol is selective for extracellular proteins, we also stained for cysteine string protein (CSP), an intracellular synaptic vesicle protein predominantly located in the presynaptic terminals (*Zinsmaier et al., 1990*; *Umbach et al., 1994*), under the same conditions. CSP was detected only in the permeabilized condition (*Figure 5E*), suggesting that the non-permeabilizing protocol is selective for extracellular proteins. We also monitored the levels of Shv-eGFP in the larval brain, confirming that the *shv-RNAi* approach successfully reduced Shv levels in the selective cell types (*Figure 5—figure supplement 1B*). Taken together, these results reveal that the amount of Shv released by neurons is homeostatically regulated by Shv produced in glia. Furthermore, unlike neurons, glial Shv release is independent of neuronal activity and does not directly alter integrin signaling.

### *Drosophila* peripheral glia and Shv control ambient extracellular glutamate levels to regulate activity-induced synaptic remodeling

If Shv derived from glia does not activate integrin signaling, how does it restore synaptic plasticity in *shv¹*? One plausible explanation is that glial Shv is required for normal glial growth or survival. We therefore monitored glial morphology when Shv is knocked down in glia by co-expressing membrane-targeted *CD4-tdGFP* using the *repo-GAL4* driver. No obvious change in overall glial morphology was observed, with glia continuing to wrap the segmental nerves and extend processes that closely associate with proximal synaptic boutons (*Figure 5—figure supplement 2*). These observations suggest that glial Shv is not essential for maintaining normal glial structure or survival and is consistent with the idea that glial Shv does not activate integrin, as integrin signaling is required to maintain the integrity of peripheral glial layers (*Xie and Auld, 2011*; *Hunter et al., 2020*).

An alternate possibility is that Shv modulates glial function to rescue activity-induced synaptic remodeling. Glial cells have an established role in providing support and maintaining glutamate homeostasis in the nervous system (**Augustin et al., 2007**). The *Drosophila* larval NMJ is a glutamatergic synapse with high ambient glutamate concentration in the hemolymph, with an average in the range of 1–2 mM glutamate (**Chen et al., 2009**). Surprisingly, the larval NMJ does not contain the excitatory amino acid transporter 1 protein (Eaat1), which is essential for removing extracellular glutamate, suggesting that high extracellular glutamate is better tolerated and removed by diffusion through the hemolymph (**Rival et al., 2006**; **Chen et al., 2009**). The high ambient glutamate concentration is maintained by the cystine/glutamate antiporter (Cx-T), which imports cystine and exports glutamate into the extracellular milieu (**Augustin et al., 2007**; **Grosjean et al., 2008**). The purpose of the high extracellular glutamate is not well understood, but it is thought to control GluR clustering and maintain a reserved pool of GluR intracellularly (**Chen et al., 2009**). Given that we observed significantly upregulated GluR clustering when Shv was reduced in glia, we hypothesized that glial Shv may influence the levels of ambient extracellular glutamate. To detect glutamate level at the synapse, we took advantage of the GAL4/UAS and LexA/LexAop systems to express the glutamate sensor (iGluSnFR) in neurons while simultaneously knocking down or upregulating Shv in glia (**Marvin et al., 2013**; **Figure 6A**). Knockdown of Shv in glia significantly reduced iGluSnFR signal, whereas upregulating Shv increased it. As a control, we also expressed Shv in neurons using *Elav-LexA* (**Figure 6A**). We found neuronal expression of Shv did not alter iGluSnFR signal at the synapse, suggesting that the change in ambient glutamate level is selectively caused by Shv from glia. Additionally, to ascertain that the decrease in iGluSnFR signal reflects a decrease in ambient extracellular glutamate levels rather than glutamate depletion caused by high levels of GluR, we upregulated GluR levels using *mhc-GluRIIA*, which drives GluRIIA expression in muscles (**Petersen et al., 1997**). We found *mhc-GluRIIA* animals exhibited elevated levels of not only GluRIIA but also the obligatory GluRIIC subunit (**Figure 6—figure supplement 1**). Despite this increase in GluR expression, we did not observe any change in extracellular glutamate levels, as measured by live imaging using the neuronal iGluSnFR sensor (**Figure 6A**). Taken together, these results suggest that glial Shv plays a critical role in controlling ambient extracellular glutamate levels.

Next, we set out to determine whether the reduced extracellular ambient glutamate is responsible for the defective synaptic remodeling observed when Shv is depleted in glia. To this end, we incubated the NMJ with HL-3 solution containing 2 mM glutamate for 1 hr before stimulating the NMJ with high KCl. Strikingly, this treatment condition not only corrected basal GluR levels, but also fully rescued the activity-induced synaptic remodeling defect seen in glial knockdown of *shv* (**Figure 6B**). In contrast, incubating the NMJ with 2 mM glutamate did not correct basal GluR or activity-induced synaptic remodeling defect when Shv is knocked down in neurons (**Figure 6—figure supplement 2**). These data are consistent with the iGluSnFR imaging results and further confirm that glial and neuronal Shv acts through distinct pathways to jointly regulate synaptic remodeling.

To further understand the function of glial cells in synaptic plasticity regulation, we ablated glia by inducing *reaper (rpr)* expression in early third-instar larvae using the inducible *repo-GeneSwitch* driver. This method allows us to examine the acute requirement for glia while avoiding lethality associated with chronic glial ablation. Following RU486 treatment for 24 hr, fragmentation of the glial cell membrane and a reduction in GFP intensity was observed (**Figure 7A**), indicating degeneration of glial cells. This is consistent with an earlier report demonstrating that Rpr expression is sufficient to induce apoptosis and trigger cell death (**White et al., 1996**). **Figure 7B** shows that glial ablation caused the same phenotype as *shv* knockdown in glial cells (**Figure 2A**), with elevated basal GluR levels and blocked activity-induced synaptic remodeling. We hypothesized that if a primary role of glial cells in synaptic plasticity regulation is to maintain ambient extracellular glutamate levels, similar to glial Shv, then correcting extracellular glutamate concentration should be sufficient to restore synaptic plasticity, even in the absence of functional glia. Strikingly, incubating NMJs with 2 mM glutamate not only restored basal GluR levels but also rescued activity-induced synaptic remodeling caused by glial ablation (**Figure 7B, C**). These results are in line with previous reports that showed low ambient glutamate levels significantly elevated GluR intensity at the *Drosophila* NMJ (**Chen et al., 2009**), but such increase can be reversed by glutamate supplementation (**Augustin et al., 2007**; **Chen et al., 2009**). Together, these data confirm that the ability of peripheral glial cells to maintain high ambient extracellular glutamate concentrations at the NMJ is crucial for synaptic plasticity.

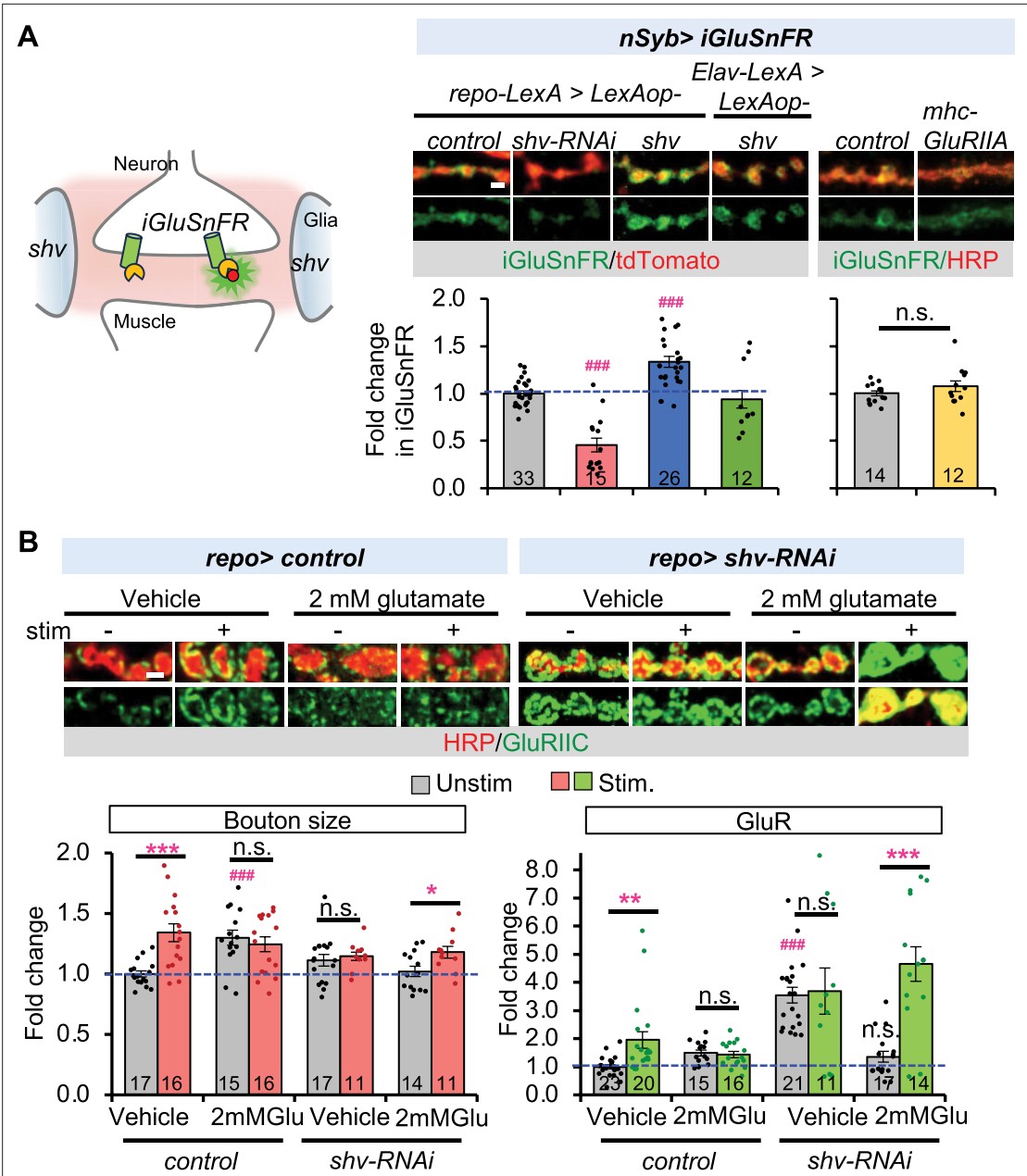

**Figure 6.** Glial Shv regulates ambient extracellular glutamate concentration. (**A**) Shv expression in glia maintains ambient glutamate concentration at the neuromuscular junction (NMJ). The left panel shows a schematic of the tripartite synapse at the NMJ. iGluSnFR expression in neurons via the GAL4/UAS system can detect extracellular ambient glutamate concentration at the synapse, while glia-specific knockdown or upregulation of Shv is achieved using the LexA/LexAop system. iGluSnFR signals are seen in green and mtdTomato marks neuronal membrane. In addition, overexpression of postsynaptic GluR using *mhc-GluRIIA* does not affect iGluSnFR signals. HRP-Cy3 was added after live imaging of iGluSnFR. Lower graph shows quantitation of the relative ambient glutamate concentration at the NMJ. Only Shv from glia affects ambient glutamate concentration, Shv from neurons does not. (**B**) Incubating the NMJ with 2 mM glutamate rescues synaptic remodeling in case of glial Shv knockdown. Vehicle controls represent NMJs dissected in parallel and incubated with HL3 without 2 mM glutamate for the same length of time. Scale bar = 2 μm. All values are normalized to unstimulated control and presented as mean ± SEM. Control contains TRiP RNAi control vector driven by the indicated driver. Statistics: One-way ANOVA followed by Tukey's multiple comparison test was used to compare between unstimulated control and unstimulated NMJs across genotypes. Student's *t*-test was used to compare between unstimulated and stimulated NMJs of the same genotype. ###p ≤ 0.001 when comparing unstimulated samples to unstimulated control. *p ≤ 0.05; **p ≤ 0.01; ***p ≤ 0.001 when comparing stimulated to unstimulated NMJs.

The online version of this article includes the following source data and figure supplement(s) for figure 6:

**Source data 1.** Data for relative iGluSnFR sensor intensity, bouton size, and GluR levels.

*Figure 6 continued on next page*

*Figure 6 continued*

**Figure supplement 1.** Validation of GluR intensity in *MhcGluRIIA*.

**Figure supplement 1—source data 1.** Data for relative levels of GluRIIA and GluRIIC subunits.

**Figure supplement 2.** Defective activity-induced synaptic remodeling caused by the loss of neuronal Shv is not rescued by incubation with 2 mM glutamate.

**Figure supplement 2—source data 1.** Data for relative bouton size and GluR intensity for the indicated conditions.

## Discussion

In this study, we show that the peripheral glial cells at the *Drosophila* NMJ play an important role in regulating synaptic plasticity. We demonstrate that neurons and glia jointly orchestrate activity-induced synaptic remodeling at the NMJ, with Shv playing a pivotal role. While neurons release Shv in an activity-dependent manner to regulate synaptic remodeling through integrin signaling (*Lee et al., 2017*), release of Shv by peripheral glial cells does not rely on neuronal activity and does not activate integrin signaling (*Figure 5A–C*). Instead, glial Shv influences activity-induced synaptic remodeling by keeping the levels of Shv released by neurons in check (*Figure 5D–F*) and controlling ambient extracellular glutamate levels (*Figure 7*).

Our data suggest that one mechanism underlying activity-induced synaptic remodeling by glia is through indirect control of Shv release from neurons, thereby maintaining minimal integrin signaling under ambient conditions. We propose that this low baseline integrin signaling enables neurons to respond with high sensitivity to enhanced integrin activation by Shv release from neurons following neuronal activity, leading to rapid synaptic remodeling. Conversely, knocking down Shv in glia removes this suppression, resulting in increased Shv release from neurons, higher basal integrin signaling, GluR levels, and pathway saturation, thereby inhibiting activity-induced synaptic remodeling. Supporting this, overexpression of Shv in neurons elevated basal pFAK and abolished synaptic plasticity (*Lee et al., 2017*). These findings also raise several intriguing questions, including how neurons and glia distinguish different sources of Shv, and how they sense and communicate this information to regulate extracellular Shv levels from different cells. Based on western blot analyses of adult heads and larval brains showing that Shv is present as a single band (*Figure 1A*, *Figure 2—figure supplement 1B*), the functional differences in neuronal or glial Shv are not likely due to the presence of different isoforms. Consistent with this, FlyBase also suggests that *shv* encodes a single isoform (*Öztürk-Çolak et al., 2024*). However, while we did not detect obvious post-translational modifications when Shv protein was expressed in neurons or glia (*Figure 5—figure supplement 1A*), we cannot exclude the possibility that different cell types process Shv differently through post-transcriptional or post-translational mechanisms. Notably, *shv* is predicted to undergo A-to-I RNA editing, including an editing site in the coding region, which will result in a single amino acid change (*St Laurent et al., 2013*). Given that ADAR, the editing enzyme, is enriched in neurons and absent from glia (*Jepson et al., 2011*), it is possible that such cell-specific editing could contribute to functional differences. It will be interesting to investigate this in the future.

We found that another main function of glial Shv is to regulate ambient extracellular glutamate concentration. We report that overexpression of *shv* elevated ambient glutamate levels at the synapse measured using a genetically encoded glutamate sensor, whereas knockdown of *shv* in glia reduced its level (*Figure 6B*). Furthermore, ablating glia recapitulated the phenotypes of *shv* knockdown in glia, and transiently restoring extracellular ambient glutamate concentration efficiently rescued synaptic plasticity. Based on reports that it has been shown that the *Drosophila* larval NMJ maintains a surprisingly high ambient extracellular glutamate concentration, with an average in the range of 1–2 mM (*Augustin et al., 2007*). How Shv regulates extracellular glutamate concentration remains to be explored, but a likely mechanism is by affecting the levels or functions of the cystine/glutamate (Cx-T), which imports cystine and exports glutamate into the extracellular matrix. While Shv has been shown to activate integrin, it encodes a homolog of the mammalian DNAJB11 protein (*Lee et al., 2016*), which functions as a molecular chaperone vital for proper protein folding in the endoplasmic reticulum (*Shen et al., 2002*). Shv thus could potentially be required for the proper folding and the function of the Cx-T, which is located on glial cell membrane at the fly NMJ (*Augustin et al., 2007*; *Grosjean et al., 2008*). Aligned with this, mutations in Cx-T also resulted in elevated basal GluR levels at the fly NMJ (*Augustin et al., 2007*). Future studies examining the functional interactions between

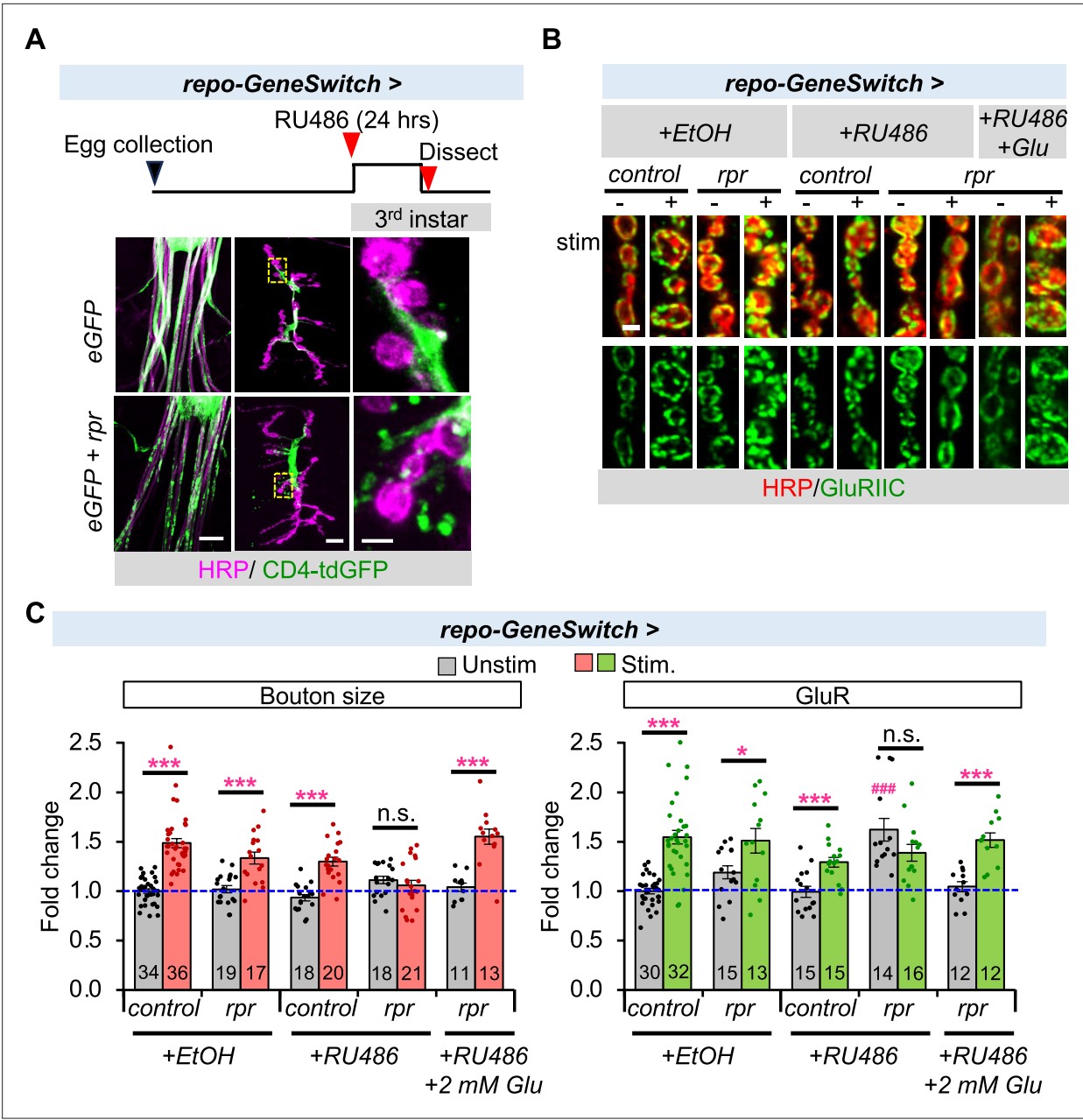

**Figure 7.** Acute ablation of glia elevated basal GluR levels and disrupted activity-induced synaptic remodeling, but extracellular glutamate incubation is sufficient to compensate for the loss of glia. (**A**) Transient *rpr* expression is sufficient to trigger death of glial cells. The diagram shows the RU486 feeding protocol used to induce glial cell death in third-instar larvae. Lower panels show images of the segmental nerves (left) and the neuromuscular junction (NMJ, middle). The zoomed-in region of the NMJ (yellow box) is magnified on the right. Glial membrane is marked by CD4-tdGFP, which appears fragmented in the presence of rpr expression, indicating glial cell death. Scale bars indicate size in 50, 10, and 2 µm from left to right. (**B**) Images and (**C**) quantitation of synaptic bouton size and GluR abundance. Incubating the NMJ for 10 min with 2 mM glutamate is sufficient to overcome loss of glial cells and restore basal GluR level and activity-induced synaptic remodeling, suggesting a main function of peripheral glia in synaptic plasticity regulation is to maintain a high ambient glutamate concentration. Scale bar = 2 µm in (**B**). All values are normalized to unstimulated control and presented as mean ± SEM. Control contains TRiP RNAi control vector driven by the indicated driver. Statistics: One-way ANOVA followed by Tukey's multiple comparison test was used to compare between unstimulated control and unstimulated NMJs across genotypes. Student's *t*-test was used to compare between unstimulated and stimulated NMJs of the same genotype. ###p ≤ 0.001 when comparing unstimulated samples to unstimulated control. *p ≤ 0.05; ***p ≤ 0.001 when comparing stimulated to unstimulated NMJs.

The online version of this article includes the following source data for figure 7:

**Source data 1.** Data for relative bouton size and GluR intensity for the indicated conditions.

Cx-T and Shv will shed light on mechanisms for Shv-dependent regulation of ambient extracellular glutamate levels at the NMJ.

How does extracellular glutamate regulate GluR levels and synaptic plasticity? Changes in glutamate levels have been shown to directly impact neurotransmission, glutamate receptor activity, and influence GluR clustering by internalizing the desensitized GluR (*Featherstone and Shippy, 2008*; *Chen et al., 2009*; *Yao et al., 2018*). We have also shown that basal GluR level is homeostatically regulated by extracellular glutamate concentration (*Figure 7*). It is plausible that a high extracellular glutamate concentration enables the NMJ to maintain a reserved pool of GluRs that can be readily mobilized to the surface upon neuronal activity. Additionally, activation of GluR and downstream signaling pathways could trigger local protein translation machineries to prime the synapses to respond rapidly to changes in neuronal activity. Future studies examining intracellular mechanisms controlling activity-induced GluR increases will lead to better insights on synaptic plasticity regulation.

# Materials and methods

## Key resources table

| Reagent type (species) or resource | Designation | Source or reference | Identifiers | Additional information |
|---|---|---|---|---|
| Strain, strain background (*Drosophila melanogaster*) | repo-GAL4 | BDSC | 7415, RRID:BDSC_7415 | Pan-glial driver |
| Strain, strain background (*D. melanogaster*) | 24B-GAL4 | BDSC | 1767, RRID:BDSC_1767 | Muscle driver |
| Strain, strain background (*D. melanogaster*) | elav-GAL4 | BDSC | 458, RRID:BDSC_458 | Pan-neuronal driver |
| Strain, strain background (*D. melanogaster*) | nSyb-GAL4 | BDSC | 51635, RRID:BDSC_51635 | Pan-neuronal driver |
| Strain, strain background (*D. melanogaster*) | Gliotactin-GAL4 | BDSC | 9030, RRID:BDSC_9030 | Subperineurial glial driver |
| Strain, strain background (*D. melanogaster*) | Nrv- GAL4 | BDSC | 6800, RRID:BDSC_6800 | Wrapping glial driver |
| Strain, strain background (*D. melanogaster*) | alrm-GAL4 | BDSC | 67032, RRID:BDSC_67032 | Astrocyte driver |
| Strain, strain background (*D. melanogaster*) | NP6293-GAL4 | Kyoto | 105188, RRID:DGGR_105188 | (*PG-GAL4*) perineurial glial driver |
| Strain, strain background (*D. melanogaster*) | TRiP-RNAi control | BDSC | 35788, RRID:BDSC_35788 | Non-specific RNAi control |
| Strain, strain background (*D. melanogaster*) | iGluSnFR | BDSC | 59611, RRID:BDSC_59611 | Glutamate concentration sensor |
| Strain, strain background (*D. melanogaster*) | UAS-CD4-tdGFP | BDSC | 35836, RRID:BDSC_35836 | Membrane form reporter |
| Strain, strain background (*D. melanogaster*) | UAS-IVS-myr::tdTomato | BDSC | 32221, RRID:BDSC_32221 | Membrane form reporter |
| Strain, strain background (*D. melanogaster*) | Elav-LexA | BDSC | 52676, RRID:BDSC_52676 | Pan-neuronal driver |
| Strain, strain background (*D. melanogaster*) | repo-LexA | Gift from Dr. Henry Y. Sun | | Pan-neuronal driver |
| Strain, strain background (*D. melanogaster*) | repo-GeneSwitch-GAL4 | *Artiushin et al., 2018* | | Drug inducible pan-glial driver |
| Strain, strain background (*D. melanogaster*) | Mhc.GluRIIA.Myc | BDSC | 64258, RRID:BDSC_64258 | GluRIIA expression in muscles |
| Strain, strain background (*D. melanogaster*) | UAS-shv-RNAi | VDRC | 108576, RRID:Flybase_FBst0480386 | |
| Strain, strain background (*D. melanogaster*) | UAS-shv-RNAi[37507] | BDSC | 37507, RRID:BDSC_37507 | |
| Strain, strain background (*D. melanogaster*) | UAS-Shv | *Lee et al., 2017* | | *shv* transgene for overexpression |

*Continued on next page*

*Continued*

| Reagent type (species) or resource | Designation | Source or reference | Identifiers | Additional information |
|---|---|---|---|---|
| Strain, strain background (*D. melanogaster*) | *shv[1]* | **Lee et al., 2017** | | *shv* mutant |
| Strain, strain background (*D. melanogaster*) | Shv-eGFP | This study | | eGFP insertion line |
| Strain, strain background (*D. melanogaster*) | Cas9 | BDSC | 55821, RRID:BDSC_55821 | CRISPR Fly injection |
| Strain, strain background (*D. melanogaster*) | LexAop-Shv- RNAi | This study | | *shv-RNAi* designed based on sequence from VDRC 108576 |
| Strain, strain background (*D. melanogaster*) | LexAop-Shv | This study | | *shv* transgene for overexpression |
| Strain, strain background (*D. melanogaster*) | P{CaryP}attP18 | BDSC | 32107, RRID:BDSC_32107 | |
| Recombinant DNA reagent | pU6-BbsI-chiRNA vector | Addgene | 45946, RRID:Addgene_45946 | |
| Recombinant DNA reagent | pHD-DsRed | Addgene | 51434, RRID:Addgene_51434 | |
| Recombinant DNA reagent | pJFRC19-13XLexAop2-IVS-myr::GFP vector | Addgene | 26224, RRID:Addgene_26224 | |
| Antibody | Rabbit polyclonal anti-pFAK | Invitrogen | Catalog #44-624G, RRID:AB_2533701 | 1:250 |
| Antibody | Rabbit polyclonal anti-HA | Sigma | Catalog #H6908, RRID:AB_260070 | 1:1000 |
| Antibody | Rabbit polyclonal anti-GluRIIC | **Chang et al., 2024** | | 1:1000 |
| Antibody | Rat monoclonal anti-Elav | DSHB | 7E8A10, RRID:AB_528218 | 1:500 |
| Antibody | Mouse monoclonal anti-Repo | DSHB | 8D12, RRID:AB_528448 | 1:50 |
| Antibody | Mouse monoclonal anti-GFP | DSHB | 4C94C9, RRID:AB_2617422 | 1:100 |
| Antibody | Cy3 conjugated anti-HRP | Jackson ImmunoResearch | RRID:AB_2338959 | 1:100 for non-permeabilized staining. 1:250 for permeabilized staining. |
| Antibody | Alexa-647 conjugated anti-HRP | Jackson ImmunoResearch | RRID:AB_2338967 | 1:100 for non-permeabilized staining. 1:250 for permeabilized staining. |
| Antibody | Alexa Fluor 488 | Invitrogen | RRID:AB_143165 | 1:250 |
| Antibody | Alexa Fluor 405 | Invitrogen | RRID:AB_221604 | 1:250 |

## Fly stocks

Flies were maintained at 25°C on a standard fly food consisting of cornmeal, yeast, sugar, and agar, under a 12-hr light/dark cycle unless otherwise specified. The following fly lines were obtained from the Bloomington *Drosophila* Stock Center at Indiana University (BDSC, stock numbers in parentheses), the Vienna *Drosophila* Resource Center and *Drosophila* Genomics Resource Center (VDRC, specified before stock number), or Kyoto stock center (Kyoto, stock number in parentheses): *repo-GAL4* (BDSC, 7415), *24B-GAL4* (BDSC, 1767), *elav-GAL4* (BDSC, 458), *nSyb-GAL4* (BDSC, 51635), *Gliotactin-GAL4* (BDSC, 9030), *Nrv- GAL4* (BDSC, 6800), *alrm-GAL4* (BDSC, 67032), TRiP-RNAi control (BDSC, 35788), *iGluSnFR* (BDSC, 59611), *UAS-CD4-tdGFP* (BDSC, 35836), *UAS-IVS-myr::tdTomato* (BDSC, 32221), *Elav-LexA* (BDSC, 52676), *UAS-shv-RNAi* (VDRC 108576), *UAS-shv-RNAi[37507]* (BDSC, 37507), *Mhc.GluRIIA.Myc (BDSC, 64258)*, *PG-GAL4* (also known as *NP6293-GAL4*, Kyoto, 105188), and, *repo-GeneSwitch-GAL4* (**Artiushin et al., 2018**), *repo-LexA* (a gift from Dr. Henry Y. Sun), *UAS-Shv* and *shv[1]* (**Lee et al., 2017**).

To visualize the endogenous levels of Shv, Shv with eGFP inserted into the 3'-end of the endogenous Shv gene (Shv-eGFP) was generated by CRISPR/Cas9 genome editing and HDR as described (**Gratz et al., 2014**). Briefly, a target Cas9 cleavage site in Shv was selected at the 3' end of Shv without obvious off-target sequence in the *Drosophila* genome using CRISPR optimum Target Finger. The sgRNA target sequence: 5'-GCCGGCGAGCACTTTTATTG was cloned into the pU6-BbsI-chiRNA vector (Addgene, #45946). The vector for HDR contains the 5' homology arm containing Shv genomic region (1000 base pairs at the 3' end of the Shv gene with the stop codon removed) plus eGFP sequence, and the 3' homology arm (1020 base pair downstream of the Cas9 cleavage site in the Shv

gene region) was cloned into pHD-DsRed (Addgene, # 51434). Both cloned vectors were injected into fly stocks containing Cas9 (BDSC 55821) and positive genome editing screen by the presence of DsRed by BestGene Inc Insertion was confirmed by sequencing and western blots.

LexAop-Shv RNAi and LexAop-Shv fly lines were generated by subcloning Shv-RNAi hairpin sequence based on VDRC Shv-RNAi design or the coding regions of Shv which contains a V5 tag into the pJFRC19-13XLexAop2-IVS-myr::GFP vector (Addgene, 26224). Both lines were inserted into the X chromosome specific position by microinjecting the plasmid into a P{CaryP}attP18 fly line (BDSC, 32107).

## RU486 feeding

*repo-GeneSwitch* embryos were collected in 3 hr windows and early third-instar larvae (48 hr after larval hatching) were fed with food containing 10 µM RU486 (1:1000, 10 mM stock) or 0.1% EtOH (vehicle control) for 24 hr before dissection (*Gatto and Broadie, 2008*).

## Immunochemistry

Dissection and stimulation of the third-instar larvae fillets were done as described (*Lee et al., 2017*). Briefly, dissected NMJs were fixed for 3 min at room temperature (RT) with Boutin's fixative for GluR staining. For all other antibodies, 4% paraformaldehyde was used (25 min at RT). The fixed samples were subsequently washed with either 0.1% Triton X-100 in PBS (PBST) or with PBS alone for a detergent-free condition. The samples were then blocked with 5% normal goat serum in either PBST or PBS, as specified. Primary antibodies used were: Rabbit anti-pFAK, 1:250 (Invitrogen, 44-624G), Rabbit anti-HA, 1:1000 (Sigma, H6908), Rabbit anti-GluRIIC, 1:1000 (*Chang et al., 2024*), rat anti-Elav, 1:500 (7E8A10, Developmental Studies Hybridoma Bank at the University of Iowa (DSHB)); mouse anti-Repo, 1:50 (8D12, DSHB); mouse anti-GFP, 1:100 (4C9, DSHB). Cy3/Alexa-647 conjugated anti-HRP, 1:100 for non-permeabilized staining or 1:250 for permeabilized staining (Jackson ImmunoResearch). For iGluSnFR imaging experiments involving *Mhc-GluRIIA*, HRP staining was done after live imaging by incubating with HRP-Cy3 for 5 min. Secondary antibodies used were Alexa Fluor 488 or 405 conjugated, diluted at 1:250 (Invitrogen). For experiments with glutamate incubation, dissected preps were incubated with 2 mM glutamate solution for 1 hr or 10 min before performing stimulation protocol as indicated. Mock controls were treated the same way but without 2 mM glutamate.

## Imaging and image analysis

Images of synaptic boutons from muscle 6/7, A2 or A3 were taken using Olympus FV3000 confocal microscope at 60X with 1.6 zoom. Images showing expression patterns were taken at lower magnifications with either 10X or 60X objectives. 40x water objectives are used when performing live imaging of iGluSnFR. Mock controls (unstimulated larvae) were always dissected, immunostained, and imaged in parallel with stimulated genotypes for each experiment using the same conditions. Type Ib boutons were analyzed to establish changes in bouton size and GluR levels. Image analyses for GluRIIC, bouton size, and extracellular staining were done using the same protocol as described (*Lee et al., 2017*).

## Electrophysiology

Third-instar larvae were dissected and immersed in a modified HL-3 solution containing NaCl (70 mM), KCl (5 mM), $MgCl_2$ (10 mM), sucrose (115 mM), HEPES (5 mM), trehalose (5 mM), and $NaHCO_3$ (10 mM), at pH 7.2, with specified concentrations of $CaCl_2$ (0.25 or 0.5 mM). Current-clamp recordings were performed on muscle area 6 in abdominal segments A2 or A3, using suction electrodes to stimulate the severed ventral nerves with a 0.3 ms stimulus duration. The recording electrode, filled with 3 M KCl and having a resistance between 15 and 40 mΩ, was used to acquire data. Data with resting potentials more hyperpolarized than –60 mV were analyzed, while datasets were excluded if resting potentials deviated by more than 10 mV during recording or if there was a sudden drop in EPSP amplitude, indicating incomplete nerve function. The experimental setup included an Axopatch 200B Amplifier, a Digidata 1440A for digitization, and pClamp 10.3 software (Molecular Devices) for control. Data analysis was conducted using MiniAnalysis (Synaptosoft), Clampfit (Molecular Devices), and Microsoft Excel, with nonlinear summation applied to correct the average EPSP.

## Western blot

*Drosophila* larval brain extract was obtained by homogenizing 6–10 larval brains collected on ice in RIPA buffer with EDTA 50 mM Tris-HCl, pH 7.5, 1% NP-40, 0.5% NaDoc, 150 mM NaCl, 0.1% SDS,

10 mM EDTA, 50 mM NaF, 1 mM $Na_3VO_4$, 250 nM cycloporin A, protease inhibitor cocktail (Roche) using mortar and pestle. Protein homogenate was separated by 10% SDS–PAGE and transferred to nitrocellulose membranes. Primary antibodies were diluted in blocking solution as follows: rabbit anti-Shv, 1:400 (*Lee et al., 2017*); Rabbit Anti-GFP, 1:1000 (NOVUS, NB600-308), Anti-β-tubulin, 1:500 (E7, DSHB), and Mouse Anti-Complex V 1:10000 (MitoScience).

## Statistical analysis

All data are shown as mean ± SEM. Sample sizes, indicated in the graphs or figure legends, represent biological replicates and adhere to established standards in the literature. Comparisons between unstimulated and stimulated samples of the paired genotype were made using Student's *t*-test. For comparisons involving multiple samples, one-way ANOVA followed by Tukey's multiple comparison test was employed to determine statistical significance. To minimize bias, all samples were randomized during dissection, image collection, and data analysis.

## Acknowledgements

We would like to thank Dr. Amita Seghal and Dr. Henry Y Sun for the generous gift of *repo-GeneSwitch-GAL4* and *repo-LexA*, respectively. KTC is supported by NIH grants R01NS102260 and R01NS080946.

## Additional information

### Funding

| Funder | Grant reference number | Author |
|---|---|---|
| National Institute of Neurological Disorders and Stroke | R01NS102260 | Karen T Chang |
| National Institute of Neurological Disorders and Stroke | R01NS080946 | Karen T Chang |

The funders had no role in study design, data collection, and interpretation, or the decision to submit the work for publication.

### Author contributions

Yen-Ching Chang, Conceptualization, Data curation, Formal analysis, Validation, Investigation, Methodology, Writing – original draft, Writing – review and editing; Yi-Jheng Peng, Data curation, Formal analysis, Writing – review and editing; Joo Yeun Lee, Data curation, Formal analysis, Methodology, Writing – review and editing; Annie Wen, Formal analysis, Writing – review and editing; Karen T Chang, Conceptualization, Resources, Data curation, Formal analysis, Supervision, Funding acquisition, Validation, Investigation, Methodology, Writing – original draft, Project administration, Writing – review and editing

### Author ORCIDs

Joo Yeun Lee ⓘ https://orcid.org/0000-0001-6616-5594
Karen T Chang ⓘ https://orcid.org/0000-0001-7619-8170

Reviewer #2 (Public review): https://doi.org/10.7554/eLife.104126.3.sa1
Reviewer #3 (Public review): https://doi.org/10.7554/eLife.104126.3.sa2
Author response https://doi.org/10.7554/eLife.104126.3.sa3

## Additional files

### Supplementary files
MDAR checklist

### Data availability
All data generated or analyzed during this study are included in the manuscript and supporting files. All data related to figures are included in the source data files.

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
