## [Editor Report · eLife Assessment]

This study presents a **valuable** finding on a new role of glia in activity-dependent synaptic remodeling using the Drosophila NMJ as a model system. The evidence supporting the claims of the authors is **convincing**. The authors have addressed most of the reviewers' concerns and help to further clarify the claims. The work will be of interest to neuroscientists working on glia-neuron interaction and synaptic remodeling.

---

## [Referee Report · Reviewer #2 (Public review)]

In this paper Chang et al follow up on their lab's previous findings about the secreted protein Shv and its role in activity-induced synaptic remodeling at the fly NMJ. Previously they reported that shv mutants have impaired synaptic plasticity. Normally a high stimulation paradigm should increase bouton size and GluR expression at synapses but this does not happen in shv mutants. The phenotypes relating to activity-dependent plasticity were completely recapitulated when Shv was knocked down only in neurons and could be completely rescued by incubation in exogenously applied Shv protein. The authors also showed that Shv activation of integrin signaling on both the pre- and post-synapse was the molecular mechanism underlying its function in plasticity. Here they extend their study to consider a role of Shv derived from glia in modulating synaptic features at baseline and remodeling conditions. The authors show evidence that Shv is expressed in both neurons and glia. Despite the fact that neuron-specific RNAi knockdown of Shv recapitulated the plasticity phenotypes seen in whole animal mutants, the authors asked whether glial-specific knockdown would have any effects. Surprisingly, knockdown of Shv only in glia also blocked plasticity, just like neuron-specific knockdown, and supporting an important role for glial-derived Shv in plasticity. Unlike neuronal knockdown, though, glial knockdown also caused abnormally high baseline GluR expression. Restoring Shv in ONLY glia in mutant animals is sufficient to completely rescue the plasticity phenotypes and baseline GluR expression, but glial-Shv does not appear to activate integrin signaling which was shown to be the mechanism for neuronally derived Shv to control plasticity. This suggests a different or indirect mechanism of action for glial-derived Shv. This led the authors to hypothesize that glial Shv might work via controlling the levels of neuronal Shv and/or extracellular glutamate. To test these hypotheses, they provide evidence that in the absence of glial Shv, synaptic levels of Shv go up overall, suggesting that glial Shv could somehow have a suppressive effect on release of neuronal Shv. This would indirectly modulate integrin signaling to control plasticity. Using an extracelluar glutamate sensor in presynaptic boutons, they also observe decreased signal (extracellular glutamate) from the sensor in glial Shv KD animals, and increased signal in glial Shv overexpression animals, supporting the hypothesis that glial Shv can regulate glutamate levels somehow. These results establish glia as an important source of Shv in these processes and identify some mechanisms for how this might be accomplished. Several outstanding questions remain-most importantly: how/why do glial-derived and neuronal-derived Shv have different effects when in the same space? No obvious isoform or size differences were found, and the same rescue construct expressed either in neurons or glia could have different effects on integrin activation or glutamate levels. Answering these questions using modified rescue constructs will be an important future direction to understand Shv function specifically and how neurons and glia work together in this context--and potentially many other contexts.

Comments on revisions:

The authors addressed my and the other reviewers' concerns from the original review adequately and this has strengthened the paper substantially.

One small omission to correct: In Figures 4 and 6, the graphs in the figures do not have a legend for the colored bars.

---

## [Referee Report · Reviewer #3 (Public review)]

Summary:

The manuscript by Chang and colleagues provides compelling evidence that glia-derived Shriveled (Shv) modulates activity-dependent synaptic plasticity at the Drosophila neuromuscular junction (NMJ). This mechanism differs from the previously reported function of neuronally released Shv, which activates integrin signaling. They further show that this requirement of Shv is acute and that glial Shv supports synaptic plasticity by modulating neuronal Shv release and the ambient glutamate levels. However, there are a number of conceptual and technical issues that need to be addressed.

Major comments

(1) From the images provided for Fig 2B +RU486, the bouton size appears to be bigger in shv RNAi + stimulation, especially judging from the outline of GluR clusters.

(2) The shv result needs to be replicated with a separate RNAi.

(3) The phenotype of shv mutant resembles that of neuronal shv RNAi - no increased GluR baseline. Any insights why that is the case?

(4) In Fig 3B, SPG shv RNAi has elevated GluR baseline, while PG shv RNAi has a lower baseline. In both cases, there is no activity induced GluR increase. What could explain the different phenotypes?

(5) In Fig 4C, the rescue of PTP is only partial. Does that suggest neuronal shv is also needed to fully rescue the deficit of PTP in shv mutants?

(6) The observation in Fig 5D is interesting. While there is a reduction in Shv release from glia after stimulation, it is unclear what the mechanism could be. Is there a change in glial shv transcription, translation or the releasing machinery? It will be helpful to look at the full shv pool vs the released ones.

(7) In Fig 5E, what will happen after stimulation? Will the elevated glial Shv after neuronal shv RNAi be retained in the glia?

(8) It would be interesting to see if the localization of shv differs based on if it is released by neuron or glia, which might be able to explain the difference in GluR baseline. For example, by using glia-Gal4>UAS-shv-HA and neuronal-QF>QUAS-shv-FLAG. It seems important to determine if they mix together after release? It is unclear if the two shv pools are processed differently.

(9) Alternatively, do neurons and glia express and release different Shv isoforms, which would bind different receptors?

(10) It is claimed that Sup Fig 2 shows no observable change in gross glial morphology, further bolstering support that glial Shv does not activate integrin. This seems quite an overinterpretation. There is only one image for each condition without quantification. It is hard to judge if glia, which is labeled by GFP (presumably by UAS-eGFP?), is altered or not.

(11) The hypothesis that glutamate regulates GluR level as a homeostatic mechanism makes sense. What is the explanation of the increased bouton size in the control after glutamate application in Fig 6?

(12) What could be a mechanism that prevents elevated glial released Shv to activate integrin signaling after neuronal shv RNAi, as seen in Fig 5E?

(13) Any speculation on how the released Shv pool is sensed?

Comments on revisions:

The authors have addressed most of my previous comments and questions in their revision.

---

## [Author Response]

The following is the authors’ response to the original reviews.

**Reviewer #1 (Public review):**
In this manuscript, Chang et al. investigated the cell type-specific role of the integrin activator Shv in activity-dependent synaptic remodeling. Using the Drosophila larval neuromuscular junction as a model, they show that glial-secreted Shv modulates synaptic plasticity by maintaining the extracellular balance of neuronal Shv proteins and regulating ambient extracellular glutamate concentrations, which in turn affects postsynaptic glutamate receptor abundance. Furthermore, they report that genetic perturbation of glial morphogenesis phenocopies the defects observed with the loss of glial Shv. Altogether, their findings propose a role for glia in activity-induced synaptic remodeling through Shv secretion. While the conclusions are intriguing, several issues related to experimental design and data interpretation merit further discussion.

We appreciate the insightful and constructive comments. We have added new data and modified the text to address your concerns. In doing so, the manuscript has been substantially strengthened. Please see our detailed point-by-point response below.

**Reviewer #2 (Public review):**
In this paper Chang et al follow up on their lab's previous findings about the secreted protein Shv and its role in activity-induced synaptic remodeling at the fly NMJ. Previously they reported that shv mutants have impaired synaptic plasticity. Normally a high stimulation paradigm should increase bouton size and GluR expression at synapses but this does not happen in shv mutants. The phenotypes relating to activity dependent plasticity were completely recapitulated when Shv was knocked down only in neurons and could be completely rescued by incubation in exogenously applied Shv protein. The authors also showed that Shv activation of integrin signaling on both the pre- and post- synapse was the molecular mechanism underlying its function. Here they extend their study to consider the role of Shv derived from glia in modulating synaptic features at baseline and remodeling conditions. This study is important to understand if and how glia contribute to these processes. Using cell-type specific knockdown of Shv only in glia causes abnormally high baseline GluR expression and prevents activity-dependent increases in bouton size or GluR expression post-stimulation. This does not appear to be a developmental defect as the authors show that knocking down Shv in glia after basic development has the same effects as lifelong knockdown, so Shv is acting in real time. Restoring Shv in ONLY glia in mutant animals is sufficient to completely rescue the plasticity phenotypes and baseline GluR expression, but glial-Shv does not appear to activate integrin signaling which was shown to be the mechanism for neuronally derived Shv to control plasticity. This led the authors to hypothesize that glial Shv works by controlling the levels of neuronal Shv and extracellular glutamate. They provide evidence that in the absence of glial Shv, synaptic levels of Shv go up overall, presumably indicating that neurons secrete more Shv. In this context which could then work via integrin signaling as described to control plasticity. They use a glutamate sensor and observe decreased signal (extracellular glutamate) from the sensor in glial Shv KD animals, however, this background has extremely high GluR levels at the synapse which may account for some or all of the decreases in sensor signal in this background. Additional controls to test if increased GluR density alone affects sensor readouts and/or independently modulating GluR levels in the glial KD background would help strengthen this data. In fact, glialspecific shv KD animals have baseline levels of GluR that are potentially high enough to have hit a ceiling of expression or detection that accounts for the inability for these levels to modulate any higher after strong stimulation and such a ceiling effect should be considered when interpreting the data and conclusions of this paper. Several outstanding questions remain-why can't glial derived Shv activate integrin pathways but exogenously applied recombinant Shv protein can? The effects of neuronal specific rescue of shv in a shv mutant are not provided vis-à-vis GluR levels and bouton size to compare to the glial only rescue. Inclusion of this data might provide more insight to outstanding questions of how and why the source of Shv seems to matter for some aspects of the phenotypes but not others despite the fact that exogenous Shv can rescue and in some experimental paradigms but not others.

We appreciate your insightful comments. We have added new data and modified the text to address your concerns. In doing so, the manuscript has been substantially strengthened. Please also see the enclosed point-by-point response.

To address the question of whether altered GluR density alone affects sensor readouts, we expressed GluR using a mhc promoter-driven GluRIIA fusion line, which increases total GluRIIA expression in muscle independently of the Gal4/UAS system. As shown in Figure 6 – figure supplement 1, mhc-GluRIIA animals exhibited elevated levels of not only GluRIIA but also the obligatory GluRIIC subunit. Despite this increase in GluR expression, we did not observe any change in extracellular glutamate levels, as measured by live imaging using the neuronal iGluSnFR sensor (updated Figure 6A). These results suggest that elevated GluR density alone does not alter iGluSnFR sensors dynamics and further support our conclusions.

In regard to the question about ceiling effect, we do not think that the lack of GluR enhancement in repo>shv-RNAi is due to a saturated postsynaptic state. This is based on results in Figure 6, which shows that GluR levels can increase up to fourfold upon stimulation in the presence of glutamate, whereas repo>shv-RNAi results in only a ~2-fold increase in baseline GluR concentration. These results suggest that the synapse retains the capacity for further upregulation.

To address the question of why exogenously applied Shv activates integrin while glial derived Shv does not, we tested whether glia and neurons could differentially modify Shv. Based on Western blot analyses of adult heads and larval brains showing that Shv is present as a single band (Fig. 1A and Figure 2 – figure supplement 1B), the functional differences in neuronal or glial Shv is not likely due to the presence of different isoforms. Consistent with this, FlyBase also suggests that shv encodes a single isoform. However, while we did not detect obvious posttranslational modifications when Shv protein was expressed in neurons or glia (Figure 5 – figure supplement 1A), we cannot exclude the possibility that different cell types process Shv differently through post-transcriptional or post-translational mechanisms. Notably, shv is predicted to undergo A-to-I RNA editing, including an editing site in the coding region, which will result in a single amino acid change (St Laurent et al., 2013). Given that ADAR, the editing enzyme, is enriched in neurons and absent from glia (Jepson et al., 2011), such cell-specific editing could contribute to functional differences. It will be interesting to investigate this in the future. We have now included this in the Discussion section.

Additionally, we have now included new data on neuronal Shv rescue of shv^1^ mutants as suggested in the updated Figure 4. Consistent with previous findings that neuronal Shv rescues integrin signaling and electrophysiological phenotypes (Lee et al., 2017), we found that it also restores bouton size, GluR levels, and activity-induced synaptic remodeling. These results support the functional contribution of neuronal Shv.

**Reviewer #3 (Public review):**
Summary:The manuscript by Chang and colleagues provides compelling evidence that glia-derived Shriveled (Shv) modulates activity-dependent synaptic plasticity at the Drosophila neuromuscular junction (NMJ). This mechanism differs from the previously reported function of neuronally released Shv, which activates integrin signaling. They further show that this requirement of Shv is acute and that glial Shv supports synaptic plasticity by modulating neuronal Shv release and the ambient glutamate levels. However, there are a number of conceptual and technical issues that need to be addressed.

We appreciate the insightful and constructive comments. We have added new data and modified the text to address your concerns. In doing so, the manuscript has been substantially strengthened. Please see our detailed point-by-point response below.

Major comments:(1) From the images provided for Fig 2B +RU486, the bouton size appears to be bigger in shv RNAi + stimulation, especially judging from the outline of GluR clusters.

Thank you for pointing this out. We have selected another image to better represent the data.

(2) The shv result needs to be replicated with a separate RNAi.

We have used another independent RNAi line targeting shv to confirm our findings (BDSC 37507). This shv-RNAi^37507^ line also showed the same phenotype, including increased GluR levels and impaired activity-induced synaptic remodeling line (new Figure 2 – figure supplement 1A).

(3) The phenotype of shv mutant resembles that of neuronal shv RNAi - no increased GluR baseline. Any insights why that is the case?

This is an interesting question. We speculate that neuronal Shv normally has a dominant role in maintaining GluR levels during development, mainly through its ability to activate integrin signaling. Consistent with this, we have shown that mutations in integrin leads to a drastic reduction in GluR levels at the NMJ (Lee et al., 2017). While we have shown that neuronal knockdown of shv elevates Shv from glia (Fig. 5E), glial Shv cannot activate integrin signaling (Fig. 5B, 5C). Additionally, high levels of glial Shv will elevate ambient glutamate concentrations (Figure 6A), which will likely reduce GluR abundance and impair synaptic remodeling (Augustin et al. 2007, Chen et al., 2009, and Figure 6B). Therefore, neuronal knockdown of Shv resulted in the same phenotype as shv^1^ mutant.

(4) In Fig 3B, SPG shv RNAi has elevated GluR baseline, while PG shv RNAi has a lower baseline. In both cases, there is no activity induced GluR increase. What could explain the different phenotypes?

SPG is the middle glial cell layer in the fly peripheral nervous system and may also influence the PG layer through signaling mechanisms (Lavery et al., 2007), therefore having a stronger effect. We have now mentioned this in the text.

(5) In Fig 4C, the rescue of PTP is only partial. Does that suggest neuronal shv is also needed to fully rescue the deficit of PTP in shv mutants?

This is indeed a possibility. We have shown that neuronal and glial Shv each contribute to activity-induced synaptic remodeling through different mechanisms. It will be interesting test this in the future.

(6) The observation in Fig 5D is interesting. While there is a reduction in Shv release from glia after stimulation, it is unclear what the mechanism could be. Is there a change in glial shv transcription, translation or the releasing machinery? It will be helpful to look at the full shv pool vs the released ones.

Thank you for the suggestion. To address this, we monitored the levels of intracellular Shv using a permeabilized preparation (we found that the addition of detergent to permeabilize the sample strips away extracellular Shv). Combined with the extracellular staining results, we can get an idea about the total amount of Shv. As shown in the updated Figure 5D, intracellular Shv levels (permeabilized) remained unchanged following stimulation, indicating that there is no intracellular accumulation and that the observed decrease in extracellular Shv is unlikely due to impaired release machinery.

(7) In Fig 5E, what will happen after stimulation? Will the elevated glial Shv after neuronal shv RNAi be retained in the glia?

Thank you for the interesting question. We agree that examining Shv distribution following neuronal activity would be highly informative. While we plan to perform time-lapse experiments in future studies to address this, we feel that such analyses are beyond the scope of the current manuscript.

(8) It would be interesting to see if the localization of shv differs based on if it is released by neuron or glia, which might be able to explain the difference in GluR baseline. For example, by using glia-Gal4>UAS-shv-HA and neuronal-QF>QUAS-shv-FLAG. It seems important to determine if they mix together after release? It is unclear if the two shv pools are processed differently.

We agree that investigating whether neuronal and glial shv pools colocalize or are differentially processed is an important future direction. We hope to examine how each pool responds to stimulation in the shv^1^ mutant background using LexA and Gal4 systems in the future

(9) Alternatively, do neurons and glia express and release different Shv isoforms, which would bind different receptors?

Thank you for the questions. We have now addressed this in the discussion and also enclosed below:

Based on Western blot analyses of adult heads and larval brains showing that Shv is present as a single band (Fig. 1A and Figure 2 – figure supplement 1B), the functional differences in neuronal or glial Shv is not likely due to the presence of different isoforms. Consistent with this, FlyBase also suggests that shv encodes a single isoform (Ozturk-Colak et al., 2024). However, while we did not detect obvious post-translational modifications when Shv protein was expressed in neurons or glia (Figure 5 – figure supplement 1A), we cannot exclude the possibility that different cell types process Shv differently through posttranscriptional or post-translational mechanisms. Notably, shv is predicted to undergo A-to-I RNA editing, including an editing site in the coding region, which could result in a single amino acid change (St Laurent et al., 2013). Given that ADAR, the editing enzyme, is enriched in neurons and absent from glia (Jepson et al., 2011), such cell-specific editing could contribute to functional differences. It will be interesting to investigate this in the future.

(10) It is claimed that Sup Fig 2 shows no observable change in gross glial morphology, further bolstering support that glial Shv does not activate integrin. This seems quite an overinterpretation. There is only one image for each condition without quantification. It is hard to judge if glia, which is labeled by GFP (presumably by UAS-eGFP?), is altered or not.

Thank you for raising this concern. To strengthen our claim, we now include additional images (Figure 5, figure supplement 2). No obvious change in overall glial morphology was observed, with glia continuing to wrap the segmental nerves and extend processes that closely associate with proximal synaptic boutons (Figure 5, figure supplement 2). These observations suggest that glial Shv is not essential for maintaining normal glial structure or survival, and is consistent with the idea that glial Shv does not activate integrin, as integrin signaling is required to maintain the integrity of peripheral glial layers.

(11) The hypothesis that glutamate regulates GluR level as a homeostatic mechanism makes sense. What is the explanation of the increased bouton size in the control after glutamate application in Fig 6?

We speculate that it could be due to a retrograde signaling mechanism activated by elevated extracellular glutamate, allowing neurons to modulate bouton morphology in response to synaptic demand. It will be interesting to investigate this possibility in the future.

(12) What could be a mechanism that prevents elevated glial released Shv to activate integrin signaling after neuronal shv RNAi, as seen in Fig 5E?

One potential mechanism is post-translational or post-transcriptional processing of Shv. Although our Western blots did not reveal differences in the molecular weight of glial vs. neuronal Shv, we cannot exclude the possibility that modifications not readily detectable by this method are responsible. Additionally, as mentioned in the Discussion section, post-transcriptional processing such as A-to-I RNA editing could introduce changes in the Shv protein, potentially altering its ability to interact with or activate integrin.

(13) Any speculation on how the released Shv pool is sensed?

The same RNA editing modification mentioned earlier or post-translational modifications in Shv may also influence how it is sensed by target cells.

**Reviewer #1 (Recommendations for the authors):**
Issues Regarding Cell Type-Specific Secretion and the Role of Shv:Extracellular Secretion of Shv:(1) The data in Figure 1 suggest that Shv is not secreted under resting conditions, challenging the proposed extracellular role of Shv. It remains unclear whether Shv secretion can be confirmed using Shv-eGFP (knock-in) following high K+ stimulation.

We apologize for not being clear. In Figure 1, Shv signals we’ve shown are from permeabilized preparation, which preferentially labels intracellular Shv. We do observe secreted Shv-eGFP following stimulation (Figure 5E), consistent with our hypothesis. However, endogenous extracellular Shv-eGFP signal is very weak, and was therefore detected using the GFP antibody and amplified with a fluorescent secondary antibody. We have now also included additional controls in Figure 5E to demonstrate the specificity of the staining.

(2) In Figure 5D, total Shv staining should be included to evaluate potential presynaptic accumulation of intracellular Shv, which may lead to extracellular secretion upon stimulation. Additionally, the representative images of glial rescue do not seem to align with the quantification data; more extracellular Shv signals were observed after stimulation.

Thank you for the comments. We monitored the levels of intracellular Shv using a permeabilized preparation (detergent treatment stripped away extracellular Shv signal). When combined with non-permeabilized extracellular staining, this approach provides insights into total Shv levels. We found no intracellular accumulation of Shv and the intracellular levels remained unchanged following stimulation (updated Figure 5D), suggesting that reduced extracellular Shv is not likely due to impaired release. Additionally, we have selected another image for glial rescue by avoiding the trachea region, which better represent the quantification data.

(3) In Figure 5E, "extracellular" Shv staining in repo>shv-RNAi samples appears localized within synaptic boutons. This raises concerns about the staining protocol potentially labeling intracellular proteins. Control experiments using presynaptic cytosolic markers are needed to confirm staining specificity.

Thank you for the thoughtful suggestion. To validate that our staining protocol is selective for extracellular proteins, we also stained for cysteine string protein (CSP), an intracellular synaptic vesicle protein predominantly located in the presynaptic terminals (Zinsmaier et al., 1990; Umbach et al., 1994), under the same conditions. CSP was detected only in the permeabilized condition (updated Figure 5E), suggesting that the non-permeabilizing protocol is selective for extracellular proteins.

(4) The study does not clarify why Shv knockdown in either perineurial glia or subperineurial glia abolishes stimulus-dependent synaptic remodeling. Does Shv secretion occur from PG, SPG, or both toward the synaptic bouton?

Thank you for raising this point. SPG is the middle glial cell layer in the fly peripheral nervous system and may also influence the PG layer through signaling mechanisms (Lavery et al., 2007). Consistent with this, we observed a stronger effect on GluR levels when SPG was disrupted compared to PG. It will be interesting to distinguish whether Shv is released by PG or SPG in the future.

(5) The possibility of an inter-glial role for Shv via integrin signaling in regulating glial morphogenesis is underexplored. The rough morphological characterization in Supplemental Figure 2 requires more detailed quantification and the use of sub-glial typespecific GAL4 drivers.

We now include additional images (Figure 5, figure supplement 2) to examine the overall glial morphology. There was no obvious change in gross glial morphology, with glia continuing to wrap the segmental nerves and extend processes that closely associate with proximal synaptic boutons when shv is knocked down in glia (Figure 5, figure supplement 2). These observations suggest that glial Shv is not essential for maintaining normal glial structure or survival, and is consistent with the idea that glial Shv does not activate integrin, as integrin signaling is required to maintain the integrity of peripheral glial layers (Xie and Auld, 2011; Hunter et al., 2020).

(6) While repo>shv rescues stimulus-dependent bouton size and GluR increases in the shv mutant (Figure 5), the interaction between neuronal and glial Shv remains unclear. Does neuronal Shv influence the expression or distribution of glial Shv?

We agree that investigating whether neuronal and glial shv pools influence each other’s expression or distribution is an important future direction. We hope to investigate this in more detail in the future using LexA-LexOp and GAL4/UAS dual expression systems.

Issues Regarding the Regulation of GluR and Perisynaptic Glutamate by Glial Shv:

(7) The methodology for iGluSnFR measurement (Figure 6A) is inadequately described. If anti-HRP staining was used to normalize signals, it suggests the experiment may have involved fixed tissue. However, iGluSnFR typically measures glutamate levels in live cells, raising concerns about the validity of this approach in fixed samples.

We apologize for not being clear about the method used to measure iGluSnFR. The original figure was generated from imaging iGluSnFR signals immediately following fixation. To address the reviewer’s concern and validate these results, we have now performed live imaging experiments using a water dipping objective to measure iGluSnFR intensity in unfixed preparations (new Figure 6A). To label synaptic boutons, we co-expressed mtdTomato using the neuronal driver, nSybGAL4. The results from the live imaging experiments confirmed our original observations that glial Shv required to control ambient extracellular glutamate levels (see updated Fig. 6A and text). Additionally, to ascertain that the decrease in iGluSnFR signal reflects a decrease in ambient extracellular glutamate levels rather than glutamate depletion caused by high levels of GluR, we upregulated GluR levels using mhc-GluRIIA, which drives GluRIIA expression in muscles (Petersen et al., 1997). We found mhc-GluRIIA animals exhibited elevated levels of not only GluRIIA but also the obligatory GluRIIC subunit. However, iGluSnFR signals at the synapse remained unchanged (Figure 6A), suggesting that elevated GluR density alone does not reduce signals. Taken together, these results suggest that glial Shv plays a critical role in controlling ambient extracellular glutamate levels.

(8) As shown in Figure 2, repo>shv-RNAi increases GluR levels before high K+ stimulation, potentially saturating postsynaptic GluR expression and precluding further increases upon stimulation.

Our data in Figure 6 show that GluR levels can increase up to four-fold upon stimulation in the presence of glutamate, whereas repo>shv-RNAi results in only a ~2-fold increase in baseline GluR concentration. These results suggest that the synapse retains the capacity for further upregulation. Thus, we do not think that the lack of GluR enhancement in repo>shv-RNAi is due to a saturated postsynaptic state, but rather reflects a requirement for glial Shv in activity-dependent modulation.

(9) Despite glial shv knockdown lowering extracellular glutamate levels, GluR levels unexpectedly increase (Figure 6B). This contradicts the known requirement for high ambient glutamate concentrations to promote GluR clustering and membrane expression (Chen et al., 2009). Furthermore, adding 2 mM glutamate reverses these increases, suggesting additional complexity in the regulation of Shv synaptic remodeling.

Thank you for the comment and the opportunity to clarify this point. While it may seem counterintuitive at first glance, our observations are in line with previous reports that showed low ambient glutamate levels significantly elevated GluR intensity at the Drosophila NMJ (Chen et al., 2009), but such increase can be reversed by glutamate supplementation (Augustin et al., 2007; Chen et al., 2009). We have revised the text to more clearly reflect this connection.

(10) If glial Shv promotes GluR expression, why does the increased extracellular Shv from neuronal shv knockdown (elav>shv-RNAi, Figure 5E) fail to elicit stimulus-dependent GluR elevation?

We speculate that this is because glial Shv does not activate integrin signaling (Figure 5B, C), and elevated glial Shv increases ambient glutamate concentration (Figure 6A), thereby reducing GluR expression (Augustin et al., 2007; Chen et al., 2009). This is indeed what we observed when shv is knocked down in neurons.

Additional Issues:(11) The type of bouton used for quantification (e.g., Ib or Is boutons) is not specified, which is critical for interpreting the results.

We apologize for not being clear. We analyzed type Ib boutons as done previously (Lee et al., 2017 and Chang et al., 2024), and have now included this information in the Methods section.

(12) The extent of Shv protein depletion in the repo-GeneSwitch system needs validation to confirm the efficacy of the knockdown.

Thank you for the suggestion. We confirmed the efficiency of acute shv knockdown by the repo-GeneSwitch system by performing Western blot analysis of dissected larval brains (Figure 2 – figure supplement 1B). Acute glial knockdown using the repo-GeneSwitch driver resulted in a 30% reduction in Shv levels, similar to the decrease observed with the repo-GAL4 driver, suggesting that the GeneSwitch driver is functional. Furthermore, knockdown of shv by the ubiquitous tubulin-GAL4 driver completely eliminated Shv protein, indicating that the RNAi construct is effective.

**Reviewer #2 (Recommendations for the authors):**
(1) General comment on statistics/data presentation: The authors employ an unusual method of using both one-way ANOVA and multiple t-test stats for the same data. Would a 2-way ANOVA be the more appropriate solution to this problem (to analyze across genotype and stimulation condition)? Also a chart in the supplementals showing all comparisons rather than just the fraction explicitly reported in the graphs would be helpful (it is not clear if no indication on significance indicates no difference or just not reported between some of the baseline levels, especially since everything is presented as ratios and in some cases this could help with data interpretation of which baseline levels are different and how they compare to other baselines and other post-stim levels). Further, there are no sample sizes given for any experiment, nor are any values of means, SD, etc ever explicitly given.

We appreciate the thoughtful suggestion. While a two-way ANOVA could be used to examine interaction effects between genotype and stimulation condition, our analysis was designed to address a specific biological question: whether each genotype, independent of baseline levels, is capable of undergoing activitydependent synaptic remodeling. To this end, we used t-tests to directly compare unstimulated vs. stimulated conditions within each genotype, allowing us to determine whether stimulation produces a significant effect in an all-or-none manner. In parallel, we applied one-way ANOVA with post hoc tests to analyze differences among baseline (unstimulated) conditions across genotypes. This approach is justified by the fact that stimulation was applied acutely and separately, and therefore the baseline values should not be influenced by the stimulated condition. Because we were not aiming to compare the extent of synaptic remodeling between genotypes, we did not use a two-way ANOVA to analyze interaction effects across all conditions.

In response to the reviewer’s suggestion, we have now added the sample number in the graphs. Additionally, in the Methods section, we include information that each sample represents biological repeats, and that data are presented as fold-change relative to unstimulated controls from the same experimental batch. This normalization is necessary, as absolute GluR intensities can vary depending on microscope settings and staining conditions.

(2) To clarify distinct roles of Shv coming from neurons vs glia it would help if the authors could include more data on the rescue of shv mutants with UAS-Shv in neurons alone. This data is never shown in the manuscript and data on what effect this rescue has on the pertinent phenotypes in this paper (bouton size and GluR staining) is not reported in the referred to 2017 paper. What this does and does not do for these phenotypes has important implications for how to interpret the glia-only rescue findings.

Thank you for the suggestion. We have now included new data on neuronal Shv rescue in shv^1^ mutants as suggested (updated Figure 4A). Consistent with previous findings that neuronal Shv rescues integrin signaling and electrophysiological phenotypes (Lee et al., 2017), we found that it also restores bouton size, GluR levels, and activity-induced synaptic remodeling. These results support the functional contribution of neuronal Shv.

(3) Figure 1C: Where are the images in the periphery taken? The morphology of the glia is odd in that "blobs" of glial membrane seemingly unattached to anything else are floating about? Perhaps these are a thin stack projection and so the connection to the main glia "stalks" are just cut off? Could a specific individual synapse be shown? Also consider HRP shown on its own so that where the actual boutons are could be more clear. It seems like both the Tomato and HRP channels are really overexposed making visualizing the morphology quite confusing. Also why not use the antibody against Shv to directly visualize expression which is more direct than a knock-in tagged version?

Figure 1C shows a single optical slice of the NMJ at muscle segment 2, selected to clearly highlight Shv-eGFP localization at a branch in close contact with the glial membrane. The glial stalk is not visible in this image because it lies in a different focal plane from the branch of interest. We have now specified this information in the figure legend. In the original figure, the HRP signal (405 channel) was oversaturated, which interfered with visual clarity. In the updated Figure 1C, we reduced the intensity of overexposed channels to better reveal the weak ShveGFP signal and fine glial processes. While we have generated an antibody against Shv, the amount is extremely limited, and hence the Shv-eGFP fusion serves as a valuable tool for visualizing subcellular localization.

(4) Do glutamate levels really rise in glia Shv KD? Although iGluSnFR signal changes could it be the high level of GluR at the synapse acting as sponges to sequester glutamate so that it can't stimulate the sensor as well? One way to test this would be to overexpress or KD GluRs in muscle in wildtype (or in the repo>Shv RNAi background) to see if that alone can modulate iGluSnfR signals?

Thank you for suggesting this important control. To address the question of whether high level GluR density alone could influence neuronal iGluSnFR sensor readouts, we expressed GluR using a mhc promoter-driven GluRIIA fusion line, which increases total GluRIIA expression in muscle independently of the Gal4/UAS system. As shown in Figure 6 – figure supplement 1, mhc-GluRIIA animals exhibited elevated levels of not only GluRIIA but also the obligatory GluRIIC subunit. Despite this increase in GluR expression, we did not observe any change in extracellular glutamate levels, as measured by live imaging using the neuronal iGluSnFR sensor (updated Figure 6A). These results suggest that elevated GluR density alone does not alter iGluSnFR sensors dynamics and further support our conclusions.

(5) The authors have some Shv constructs that can't be secreted or can't bind to integrins. Performing cell type specific rescues with these constructs might also help distinguish how source matters for each proposed sub-function of Shv though this may be outside the scope of this study.

Thank you for noticing the Shv constructs we have. We hope to further test subfunctions of Shv in the future.

(6) At one point the authors discuss experiments that measure how much Shv is released by glia during neuronal stimulation. Then state that "These data indicate that glial Shv does not directly inhibit integrin signaling." But how this experiment relates to integrin signaling is not explained and unclear.

We apologize for the confusion. We have now updated the text to better explain our logic: “This activity-induced decrease in glial Shv levels, along with reduced integrin activation (Fig. 5B), suggest that glial Shv does not act by directly inhibiting integrin signaling.”

**Reviewer #3 (Recommendations for the authors):**
Minor comments(1) Readers are left wondering what causes the increased baseline of GluR after glial shv RNAi at Fig 1, which is addressed much later. It would be helpful to preemptively mention this.

Thank you for the suggestion. To maintain a logical flow, we chose to first present the phenotypic data in Figures 1 and 2 and then return to the mechanistic explanation once we introduced ambient glutamate measurements.

(2) Be consistent with eGFP vs EGFP.

Thank you, we have corrected the inconsistencies.

(3) Scale bar for Fig 1B is missing in the low-magnification panel.

Thank you for pointing out. We’ve put in the scale bar for Figure 1B.

(4) Fig 1C, it would be helpful to elaborate on the anatomy. For example, what NMJ/abdominal segment is this? Why only some axons are surrounded by glia?

Figure 1C presents a single optical slice of the NMJ at muscle segment 2, chosen to highlight Shv-eGFP localization at a branch closely juxtaposed to the glial membrane. The glial stalk is not shown in this image because it resides in a different focal plane than the branch being visualized. We have now included this information in the figure legend.

(5) For Fig 3B, while it is stated that "we observed normal synaptic remodeling using alrmGAL4," the effect size is smaller. There seems to be a decrease in the amount of synaptic remodeling occurring?

Thank you for pointing this out. Our primary goal was to determine whether each genotype, regardless of baseline GluR levels, is capable of undergoing activitydependent synaptic remodeling in response to stimulation. For this reason, we focused on detecting the presence or absence of remodeling rather than comparing the extent of remodeling across genotypes. While a smaller effect on activity-induced bouton size was observed with alrm-GAL4, the change was still statistically significant, indicating that remodeling does occur in this genotype. Currently, we do not have a clear biological interpretation for differences in the magnitude of remodeling, and therefore chose not to emphasize cross-genotype comparisons.